# In silico genomic surveillance by CoVerage predicts and characterizes SARS-CoV-2 variants of interest

Katrina Norwood [1,2,7], Zhi-Luo Deng[1,2,3,7], Susanne Reimering[1,2,7], Gary Robertson[1,2], Mohammad-Hadi Foroughmand-Araabi[1,2], Sama Goliaei[1,2,3], Martin Hölzer [4], Frank Klawonn [5,6] & Alice C. McHardy [1,2,3] ✉

Rapidly evolving viral pathogens such as SARS-CoV-2 continuously accumulate amino acid changes, some of which affect transmissibility, virulence or improve the virus' ability to escape host immunity. Since the beginning of the SARS-CoV-2 pandemic, multiple lineages with concerning phenotypic alterations, so-called Variants of Concern (VOCs), have emerged and risen to predominance. To optimize public health management and ensure the continued efficacy of vaccines, the early detection of such variants is essential. Therefore, large-scale viral genomic surveillance programs have been initiated worldwide, with data being deposited in public repositories in a timely manner. However, technologies for their continuous interpretation are lacking. Here, we describe the CoVerage system (www.sarscoverage.org) for viral genomic surveillance, which continuously predicts and characterizes emerging potential Variants of Interest (pVOIs) from country-wise lineage frequency dynamics, together with their antigenic and evolutionary alterations utilizing the GISAID viral genome resource. In a comprehensive assessment of VOIs, VUMs, and VOCs, we demonstrate how CoVerage can be used to swiftly identify and characterize such variants, with a lead time of almost three months relative to their WHO designation. CoVerage can facilitate the timely identification and assessment of future SARS-CoV-2 variants relevant for public health.

In early 2020, infections with a previously unknown coronavirus of probable zoonotic origin in Wuhan, China, were first reported[1]. The virus, named SARS-CoV-2, rapidly spread across the globe, causing over 675 million infections and 6.8 million deaths as of March 2023[2]. SARS-CoV-2 has a single-stranded, positive-sense RNA genome with substantial capacity to mutate, reflected in its strain-level diversity of circulating viral lineages and rapid evolution[3,4]. This led to the emergence of several Variants of Concern (VOCs), as designated by the World Health Organization (WHO), with altered phenotypes in transmissibility, virulence, or antigenicity, causing large waves of new infections or reinfections[5–8].

Generally, viral pathogens such as human influenza and severe acute respiratory syndrome coronavirus 2 (SARS-CoV-2) viruses evolve rapidly, adapting to the human host for efficient replication and spread. Continuous changes on the surface antigens of these viruses allow them to evade host immunity developed through either prior

[1]Computational Biology of Infection Research, Helmholtz Centre for Infection Research, Braunschweig, Germany. [2]Braunschweig Integrated Centre of Systems Biology (BRICS), Technische Universität Braunschweig, Braunschweig, Germany. [3]German Center for Infection Research (DZIF), partner site Hannover Braunschweig, Braunschweig, Germany. [4]Genome Competence Center (MF1), Robert Koch Institute, Berlin, Germany. [5]Biostatistics, Helmholtz Centre for Infection Research, Braunschweig, Germany. [6]Department of Computer Science, Ostfalia University of Applied Sciences, Wolfenbuettel, Germany. [7]These authors contributed equally: Katrina Norwood, Zhi-Luo Deng, Susanne Reimering. ✉e-mail: amc14@helmholtz-hzi.de

infection from previous strains or from vaccination. This capacity of a virus, known as immune escape, allows the virus to reinfect individuals and, consequently, vaccines protecting against such viruses need to be frequently updated to maintain their effectiveness against circulating variants[9]. SARS-CoV-2 in particular demonstrates an increased capacity for immune escape, with more recent circulating variants evading vaccine-derived antibodies and convalescent sera[10]. The Omicron variant was designated a VOC by the WHO in November 2021, and its sublineages BA.2, BA.4, and BA.5 have reduced susceptibility to monoclonal antibodies (mAbs) in clinical use[11,12]. A later Omicron subvariant, JN.1, which has rapidly risen to predominance in January 2024 with a global prevalence of 72.89%, has over 30 amino acid mutations occurring on the spike protein, including the additional mutation L455S and significantly enhanced immune escape compared to its parent lineage BA.2.86[13,14]. This increasing capacity for immune evasion is driven by key mutations throughout the spike protein, which reduce neutralization by antibodies or T-cell-based responses[10].

To monitor SARS-CoV-2 evolution and adaptation as well as enable the timely identification of new VOCs, many countries implemented large-scale viral genomic surveillance programs, leading to the generation of unprecedented amounts of sequence data. As of March 2024, more than 16.5 million sequences were available in the GISAID database[15]. Though web-based platforms offer various analyses based on publicly available sequencing data that support scientists, public health officials, and the general public in making sense of these highly complex data, there remains a need for methods that identify antigenically altered lineages among the numerous circulating lineages, particularly among lineages rapidly rising in frequency, to support

public health related decision making in a timely manner. The early identification of such variants is particularly relevant for vaccine updates to ensure continued vaccine efficacy, such as in the case of Omicron, XBB.1.5, and JN.1[16].

Here, we describe CoVerage (https://sarscoverage.org), an analytics platform that monitors the genetic and antigenic evolution of human SARS-CoV-2 viruses. The platform implements methods that continuously search for pVOIs from global, country-wise viral variant frequency dynamics and predict relevant evolutionary changes and their antigenic alterations relative to the originally described SARS-CoV-2 strain, using publicly available viral surveillance and genome data in a fully automated, timely fashion. The term pVOI, adapted from the WHO-defined Variant of Interest (VOI), defines potential Variants of Interest (pVOIs) as predicted by CoVerage, which identifies pVOIs as those that increase significantly in frequency over time and rise above a predominance threshold for the first time in a fully automated fashion[17]. We demonstrate the application of the framework for the early detection of circulating Variants of Interest (VOIs), Variants Under Monitoring (VUMs), and VOCs.

## Results
### CoVerage implementation
The CoVerage analytical workflow comprises several stages: (1) input data acquisition and filtration, (2) computational sequence data analyses, (3) creation of template pages based on a bootstrap framework, and (4) visualization in the browser using GitHub Pages (Fig. 1). Both genomic sequences and the corresponding metadata file, which contain amino acid substitutions and deletions for each given isolate as

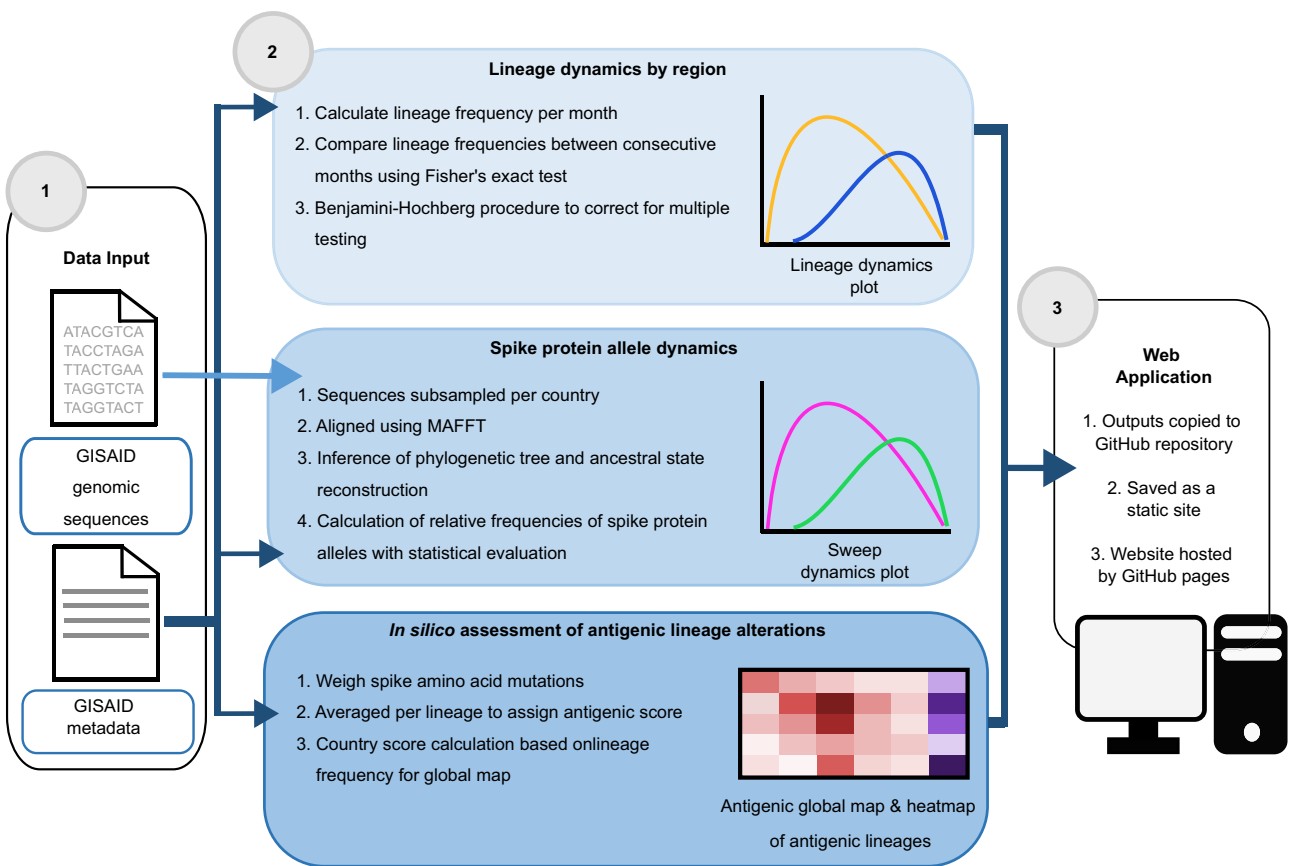

**Fig. 1 | Continuous viral genome analytics provided by the CoVerage platform (sarscoverage.org).** Data input includes GISAID genomic sequences and metadata for SARS-CoV-2 isolates from GISAID. These inputs are then fed through three computational workflows, whose outputs are saved to individual GitHub

repositories for each analysis. The results are then copied to the main CoVerage GitHub repository, where the individual pages are saved as static sites, which are hosted by GitHub Pages. Workflows are run once a week, and result pages are updated accordingly.

well as Pango lineage information, are used as inputs for the CoVerage computational analytics pipeline. For this purpose, genomic sequences and the sequence metadata file are downloaded with daily updates from GISAID[18]. Optionally, German sequence data is downloaded from Zenodo, where it is made available by the Robert Koch Institute (RKI) (https://zenodo.org/record/8334829), prior to its submission to GISAID. Alternative to GISAID metadata, the RKI metadata contains both sequence lineage assignment and submission date information from genomic surveillance efforts within Germany. Case numbers are obtained from the WHO Coronavirus (COVID-19) data repository for the global case map on the homepage, which provides an overview of cases and sequences available per country (Supplementary Fig. 1)[19]. The computational workflow performs the analysis of SARS-CoV-2 lineage dynamics by country to identify emerging lineages that may possess a selective advantage, analysis of allele dynamics of the SARS-CoV-2 major surface protein to identify alleles with amino acid changes that may provide a selective advantage, and the in silico assessment of antigenic lineage alterations. All of the analyses, the antigenic alteration analysis, the lineage dynamics analysis, and the spike protein allele dynamics analysis, require the GISAID metadata as input, though the spike protein allele dynamics also requires genomic sequences. Each of these analyses and data downloads is run independently once per week, and the results are updated on the web server accordingly.

## Lineage and spike protein dynamics analyses provided by CoVerage

CoVerage provides lineage and spike protein dynamics analyses for all countries with sufficient data available (set to more than 2000 sequences) to which one can navigate to via the interactive world map (Supplementary Fig. 1). The lineage dynamics analysis suggests potential Variants of Interest for individual countries (Fig. 1), as lineages rising in prevalence more rapidly than expected by chance (Methods), which may be due to a selective advantage and increased fitness to spread relative to other circulating viral lineages. Furthermore, CoVerage provides an analysis of spike protein allele dynamics, which identifies "lineage alleles", corresponding to branches in a viral spike protein genealogy and associated amino acid changes increasing significantly in frequency over time, which can indicate that these confer a selective advantage to the respective lineages (Methods)[20]. For the analysis of lineage dynamics and to identify pVOIs, i.e., lineages that may have a selective advantage to spread in the host population, as well as for identifying sets of amino acid changes on the spike protein rapidly increasing in frequency, we adapted a technique that we developed for recommending vaccine strain updates for the seasonal influenza vaccine (https://github.com/hzi-bifo/SDplots)[20]. There are three methodologies for identifying pVOI lineages: the "standard" method identifies lineages increasing significantly in frequency over time with monthly intervals using Fisher's exact test. The sub-lineage corrected method for pVOI identification includes genome sequence assignments belonging to sublineages in the pVOI assessment. Finally, the sliding window method uses a windowed time period for the analysis to achieve a more fine-grained analysis of shorter time intervals (Methods).

For the lineage dynamics analyses, Pango lineages were used as nomenclature, which define epidemiologically relevant phylogenetic clusters, in which new lineages are only designated if the lineage has high coverage and contains a sufficient number of sequences[21]. All methods utilize a one-sided Fisher's exact test to identify lineages that are significantly increasing in frequency over two consecutive time intervals and correct for multiple testing using the false discovery rate (FDR, $\alpha = 0.05$)[22]. This identifies pVOIs as lineages that are both significantly on the rise and increase above a predominance threshold of 0.1 within the same time frame. Notably, lineages may also be falsely identified as pVOIs due to unrepresentative sampling, e.g., data biases towards certain areas, large clonal outbreaks, or population

bottlenecks. Therefore, identified pVOIs should be evaluated carefully, e.g., in combination with epidemiological data and experimental evidence showing that amino acid changes and altered positions observed in a pVOI lineage are likely to confer a selective advantage.

## In silico assessment of antigenic lineage alterations

CoVerage identifies antigenically altered lineages using a method that scores evolutionary changes based on a viral immune escape matrix (Methods). The underlying idea of scoring amino acid changes to predict the antigenic alteration of SARS-Cov-2 lineages is that there are commonalities of how the major antigen of these viral pathogens evades the humoral immune responses evoked by the host, and that this is reflected in antigenic alteration weights for such changes that we inferred from genetic and antigenic data of seasonal influenza A viruses[10,23]. To evaluate associations between these inferred antigenic alteration weights per amino acid change and the associated physicochemical alterations in charge, a Wilcoxon rank sum test was used. The results showed that charge changes from neutral to positive and the reverse positive to neutral, as well as negative to positive and the reverse positive to negative, were associated with significantly higher antigenic weights compared to changes that did not affect charge ($p$ values of 0.018 for both charge changes, after Benjamini-Hochberg correction for multiple testing; Supplementary Fig. 2a), demonstrating that immune evasion in human influenza A/H3N2 viruses is linked to changes in charge. Indeed, amino acid changes, particularly those involving a positive charge, on the hemagglutinin (HA) protein of Influenza A and the spike protein of SARS-CoV-2 reduce the strength of interactions with neutralizing antibodies and affect binding to host receptors[24,25].

Amino acid changes assigned large antigenic alteration weights for human influenza A/H3N2 viruses are also preferentially found among antigenicity-altering changes reported for SARS-Cov-2 lineages (two-sided Fisher's exact test $p$ value = 2.88 × 10−16, Odds Ratio = 15.722, $n$ = 380; these changes referred to can be found here: https://tinyurl.com/3rk3pmzy), and these antigenic alteration weights for amino acid changes resulted in an Area under the Curve (AUC) value of 0.802 for detecting known immune escape changes of SARS-Cov-2 (Supplementary Fig. 2c), both highlighting a strong relationship in how specific amino acid changes influence antigenicity across both viruses. Thus, amino acid changes in the spike protein are weighed utilizing these viral immune escape matrices, as the spike protein plays a critical role in SARS-CoV-2's binding to host receptor cells and is a key target for vaccines and antibody treatments[26].

On CoVerage, these antigenic alteration scores are visualized in a monthly global map depicting antigenically altered circulating lineages per country and as a heatmap with antigenically altered lineages and their corresponding frequency. To identify significantly antigenically altered lineages, a z-score standardization is applied to the circulating lineages of the month, where lineages with a standardized score greater than one standard deviation from the mean are denoted as altered compared to the other circulating lineages for that month (Methods). In the antigenic alteration heatmap, significantly antigenically altered lineages are indicated with an asterisk by the lineage name, otherwise, antigenically altered lineages are ranked by frequency.

## Validation of antigenic alterations scores with geometric mean fold reduction in neutralization values of monoclonal antibodies (mAbs) and antigenic distance

To assess antigenic alteration predictions for SARS-CoV-2 lineages, we performed a thorough benchmarking versus experimentally measured antigenic cartography distances[27] and averaged mean fold reduction in neutralization (mFRN)[11]. Here we considered a comprehensive set of previously circulating and more recent lineages, including Alpha, Beta,

Delta, Gamma, Epsilon, Lambda, Mu, Omicron, and subsequent Omicron sublineages.

Neutralization assays allow systematic comparisons of the ability of viral variants, or lineages, relative to the wild-type virus, regarding their ability to infect cells in the presence of a monoclonal antibody. The fold reduction in neutralization (FRN) is determined by quantifying the concentration of a monoclonal antibody required to prevent infection of cells by a virus or pseudovirus with the spike protein of a particular SARS-CoV-2 variant and comparing these to the results for the wild type sequences, such as the originally described SARS-CoV-2 (GenBank: MN908947.3) reference sequence using the same experimental conditions. This allows for the comparison of studies even when using different experimental protocols[11]. Alternatively, the antigenic cartography distances were derived from both human and hamster sera and measured the antigenic distance between the D614G variant and a selected number of VOCs, reflective of the antigenic difference between the two strains[27].

We analyzed different ways of scoring the antigenic alterations of lineages based on their changes relative to the original SARS-CoV-2 isolate (GenBank: MN908947.3) and assessed their value using experimental data. The evaluated scoring methods include (1) scoring antigenic alterations at amino acid sites of the spike protein using M_ant_alt_0, which consists of amino acid sites identified from literature through 30 June 2021 that had a demonstrated capacity to alter the viral antigenic phenotype in the early phase of the pandemic (Supplementary Table 1). These sites were largely identified and characterized from the Alpha, Beta and Delta variants[28,29], (2) scoring antigenic alterations of directed changes across all amino acid sites of the spike protein, (3) simply counting amino acid changes that occur at the sites defined in method one, (4) counting amino acid changes across the spike protein, (5) using an alternative bidirectional scoring matrix, M_ant_alt_1, with antigenic weights averaged across change and reverse change, respectively on all altered sites for the spike protein, and finally, (6) using an alternative scoring matrix, M_ant_alt_2, which included averaged antigenic weights from the Influenza A antigenic tree for amino acid changes that occur at least three times across the tree (Methods).

Overall, each of the methods correlated strongly with both the antigenic distances and averaged mFRN values, which were used as a proxy measure of immune escape capacity in this comparison (Supplementary Table 2). Antigenic alteration scores increased between earlier lineages (Alpha, Beta, Epsilon, Lambda, Delta and Gamma) to Omicron and its subsequent lineages, as Omicron has been proven to be antigenically distinct from prior lineages with an increased immune escape capacity from sera of patients infected with a strain of the ancestral Pango lineage B[30]. Only two methods (method 2 and 5, Fig. 2c, d, i, j) after a Spearman's correlation test have a greater correlation with antigenic cartography distances and mFRN values than the baseline method, considering counts of changes across the spike protein (method 4, Fig. 2g, h; and Supplementary Table 2). With this, we selected the method that applies unidirectional immune escape weights for all amino acid changes across the spike protein (Fig. 2c, d) to progress with, as this method was significantly closer to the antigenic distances in comparison to the baseline method as per a one-sided Wilcoxon signed-rank test on the absolute deviations calculated by subtracting the normalized (using min-max normalization) scoring values from the normalized antigenic cartography distances ($n = 19$, $p$ value $= 0.0005$, Methods, Supplementary Table 3)[31]. Utilizing the normalized deviations allows for the comparison of how closely the tested method correlates to the ground truth antigenic distances in comparison to the baseline method. Additionally, the antigenic impact of an amino acid change is not symmetric; for instance, physicochemical properties of the amino acid changes are not the same bidirectionally, and some amino acid changes might alter glycosylation at specific residue sites,

while the reverse may not produce the same effect[32]. This unidirectional effect is more reflected in the chosen model.

Notably, limiting the set of sites to those defined in literature in the earlier stages of the pandemic, was not sufficient to cover the continued antigenic evolution of human SARS-CoV-2 lineages, scoring Omicron and several subsequent lineages, XBB and EG.5, lower than their antigenic cartography distances rank (Fig. 2e, f). The XBB lineage has the N460K amino acid change, which reduces neutralization, but is not scored with this model[33,34]; the EG.5 lineage carries the F456L change, as also seen in the KP.2 lineage, which increases neutralization resistance and is not scored[35,36]. Ultimately, the antigenic weights serve as an important addition to the scoring method, while the restriction to known antigenic sites does not improve the scoring.

We also compared this scoring methodology with the scoring results of both EVEscape and SpikePro, alternative viral fitness scoring programs, to antigenic distances available from the Wang et al., 2024 preprint, as well as averaged mean fold reduction in neutralization (mFRN) values for known VOC's provided by Cox et al., 2022[11,27,37,38]. EVEscape is a deep learning model that combines fitness predictions with biophysical and structural information to quantify the viral escape potential of SARS-CoV-2 mutations and strains, while SpikePro scores potential fitness of SARS-CoV-2 lineages based on predicted host ACE-2 receptor cell binding affinity, spike protein stability, and mAb binding affinity (Supplementary Fig. 3)[37,38]. When these averaged EVEscape scores were compared alongside median antigenic alteration scores calculated by CoVerage from January 2020 to December 2023 to both the antigenic cartography distances as well as averaged mFRN values for known VOC's using Spearman's correlation, the antigenic scores produced by CoVerage had a higher rho correlation for both (Supplementary Fig. 3a and Supplementary Table 4). Specifically, the median CoVerage antigenic alteration scores had a rho 0.019 and 0.1 greater for the antigenic cartography distances and averaged mFRN values, respectively and correlated significantly closer with the ground truth antigenic distances in a one-sided Wilcoxon signed-rank test (Supplementary Table 5; $p$ value $= 0.00529$, $n = 19$), serving as evidence of the antigenic alteration scoring methodology providing even more accurate antigenicity estimates than the recent EVEscape technique. The antigenic scores produced by CoVerage also had a higher Spearman's rho correlation with the antigenic distances than SpikePro (difference of 0.051, Supplementary Fig 3b and Supplementary Table 6). In a one-sided Wilcoxon signed-rank test to compare the deviations of the normalized scores of both methods to the normalized antigenic ground truth distances there was no significant difference (Supplementary Table 7), possibly due to the small sample size ($n = 11$, the number of variant scores available for SpikePro), with differences observed for individual lineages. SpikePro correctly scored Delta higher than Alpha (with scores of 6.1 and 6.9, respectively), while conversely, CoVerage correctly scored EG.5.1 higher than the XBB.1.5 lineage (21.15 and 19.61, respectively).

## Widely spreading antigenically altered SARS-CoV-2 lineages can be detected early

For validation of the pVOI identification and antigenically altered scoring methodologies created by CoVerage, we determined, for pVOIs and antigenically altered lineages predicted until the end of June 2024, the time it took for these lineages to reach their peak frequencies (Fig. 3). The frequency for each lineage in each country was calculated using a one week sliding window with a step size of 2 days. From here, the peak sequence count, the corresponding frequency, and the date of the peak were identified. To ensure robustness and mitigate bias from rare lineages or inadequate sequence sampling, we included only those records where the peak sequence count was at least 5 within the respective window. In our comprehensive evaluation, a total of 404 detected pVOIs from 91 countries were analyzed, corresponding to 1360 pVOI predictions for

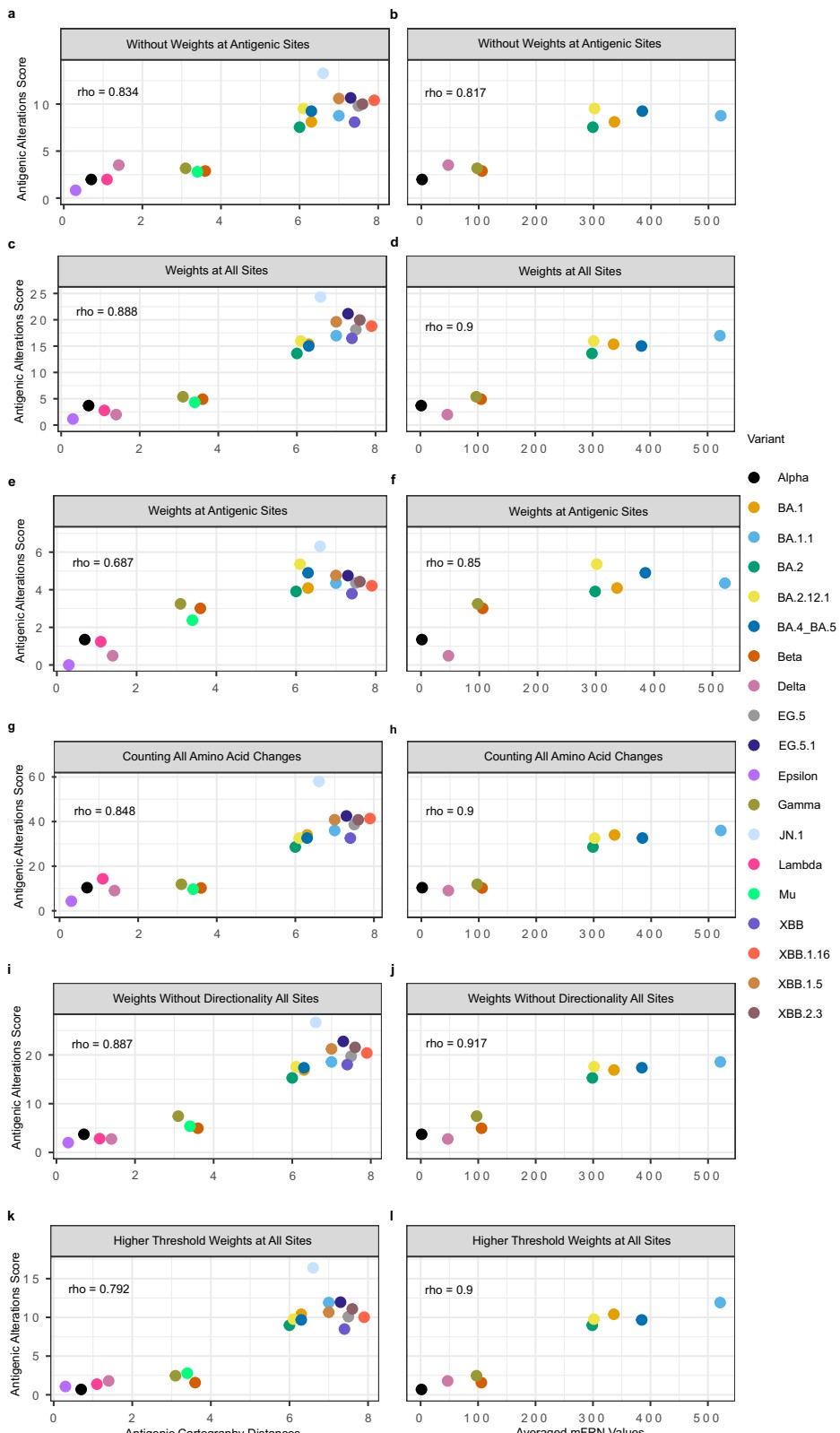

individual countries. On average, CoVerage identified a pVOI using data from 49 days earlier than the date when it reached its peak frequency. For the countries with more data available (peak sequence count ≥100 for the respective lineage), this lead time extended to 78 days (Fig. 3a). The antigenically altered pVOIs have a day difference of 72 compared to 24 for non-altered pVOIs and reached a much higher peak frequency than the non-altered ones

(Fig. 3a), indicating the relevance of antigenic alterations for providing a selective advantage to variant lineages.

Overall, 320 pVOI lineages identified by CoVerage belong to the 43 officially designated VOCs, VOIs, and VUMs, covering 31 of these designated lineages, resulting in a precision of 79% (320/404) and a recall of 72% (31/43). Among the 12 lineages not captured by pVOIs, 10 are VUMs, thus the lowest category of relevance from public health

**Fig. 2 | Assessment of lineage antigenic alteration scoring methods.** Comparison of predicted lineage antigenic alteration scores (AAS) versus experimentally determined distances from antigenic cartography ($n = 19$) and mean mFRN values for all mAbs ($n = 9$). The y-axes represent the predicted antigenic alteration scores of variants relative to the originally described SARS-CoV-2 isolate (GenBank: MN908947.3) strain using data across the spike protein from January 2020 to December 2023 (calculated from a total of 16,204,027 sequences and with arbitrary units). The x-axes represent either the antigenic cartography distance[27] or the mean mFRN values[11] respectively (with arbitrary units), with their correlation assessed using Spearman's correlation coefficient 'rho'. Antigenic alteration scores were calculated: as the number of changes at defined antigenic sites, compared to **a** antigenic cartography distances and **b** mean mFRN values; with antigenic weights at all altered sites on the spike protein compared to c antigenic cartography distances and **d** mean mFRN values; with antigenic weights at defined antigenic sites (Supplementary Table 1) on the spike protein compared to **e** antigenic cartography distances and **f** mean mFRN values; without weights at all sites on the spike protein compared to **g** antigenic cartography distances and **h** mean mFRN values; using weights at all sites but without directionality of the amino acid change compared to i antigenic cartography distances and **j** mean mFRN values; and averaging over antigenic weights from the Influenza A antigenic tree that occurred at least three times at all altered sites compared to **k** antigenic cartography distances and **l** mean mFRN values. Source data are provided as a Source Data file.

monitoring. No VOCs were missed. On average, these pVOIs were first identified using data collected 79 days before their first official WHO designation. The pVOIs not designated by WHO were primarily identified in 2020, during the early stages of the pandemic, when available sequence data was oftentimes limited and less from systematic representative surveillance efforts, as in the following years (Fig. 3b). The standardized antigenic scores of WHO-designated lineages were substantially higher than those of the non-designated lineages (Wilcoxon rank sum test $p$ value $= 1.25 \times 10^{-22}$), and increasing in order of VUMs, VOIs and VOCs, in line with their proven relevance for public health decision making (Fig. 3c). Notably, CoVerage assigns higher scores to early emerging lineages that exhibit substantial antigenicity changes compared to previously circulating variants, whereas sublineages with minor antigenic alteration from their parent lineages receive relatively lower scores. The scores of subvariants of WHO-designated lineages are lower than those of their parent lineages, although they remain higher than those of non-designated lineages. By prioritizing early emerging lineages with significant antigenic drift, this scoring approach identifies early variants that evade pre-existing immunity and warrant vaccine updates to maintain effective protection. To evaluate the benefits of our method relative to a sequence base method, we compared PyR0[39] relative to the antigenic scoring method (Fig. 3d), which showed that the antigenic alteration score from CoVerage predicts designated lineages more accurately than the $R/R_A$ score from PyR0, with an Area Under the Precision-Recall Curve (PRAUC) of 0.86 vs 0.79, respectively. Overall, these findings underscore our method's capability to effectively predict the ascent of lineages relevant for public health decision making with a growth advantage well ahead of their peak frequency, providing critical lead time for public health interventions.

## Case study 1a: CoVerage lineage dynamics allow for timely identification of Omicron as pVOI

We retrospectively analyzed data from the time of emergence of the Omicron lineage (Pango lineage BA.1), to assess the detection of pVOIs, prediction of antigenic alterations (Case study 1b), and of relevant amino acid changes (Case study 1c). This was after finalizing the pVOI detection method in the first year of the pandemic, and utilizing an antigenic alteration scoring method not including knowledge about relevant sites for Omicron or other variants. Using viral genome information provided until November 23rd of 2021, the day when the first Omicron sequences from viral genomic surveillance in South Africa[40] were submitted, CoVerage lineage dynamic analysis suggested BA.1 as a pVOI, which occurred with relative frequency of 0.27 among South African isolates (Fig. 4a). This was only 3 days prior to the swift designation of Omicron as a VOC by the WHO[41], owing to the highly efficient work and data release of South-African scientists. The respective isolates were sampled between November 14th and November 16th, demonstrating how novel lineages of concern can be rapidly identified this way when data is made public in a timely manner (Fig. 4). In the global heatmap visualizing the relative frequencies of all pVOIs detected by CoVerage lineage dynamics analysis in countries worldwide, Omicron was

identified in November 2021 in South Africa (Fig. 4c), and then rose rapidly in frequency together with multiple sublineages such as BA.1.1, BA.1.15 and BA.1.17 to the most prevalent lineage worldwide by December 2021 (Fig. 4d).

Earlier, CoVerage identified Beta (B.1.351) in September of 2020 and several Delta sublineages (AY.45, AY.38, and AY.32) throughout June and August of 2021 in South Africa as pVOIs (Fig. 4a), which were later designated VOCs by the WHO[42]. Furthermore, lineages C.1 and B.1.1.54 were identified in June and July of 2020 in the standard and sublineage-corrected analyses, which at these times rapidly increased in frequency, such that by August 2020, the C.1 lineage was the geographically most widespread lineage in South Africa[40].

## Case study 1b: spike protein allele dynamics suggest emerging lineages and their associated amino acid changes in South Africa

For South Africa, CoVerage detected the Beta lineage via a set of associated changes in the spike protein in October 2020, as well as Delta in May 2021 (Fig. 4b). Key identified substitutions of the Beta lineage include the deletion at position 241–243, which reduces binding of certain nAbs, and K417N, which is also found in Alpha[43]. K417N, in combination with E484K in Beta, reduces mAb neutralization, but does not display this capacity alone[11]. Key phenotype-altering changes of the Delta lineage identified in allele dynamics include L452R, which is an important driver of adaptive evolution; T478K, which improves immune escape; and P681R, which enhances furin cleavage[43,44]. In November 2021, a lineage allele including all 35 amino acid changes representative of BA.1 was detected (Fig. 4b). Among the changes altering the phenotype and providing a selective advantage are N501Y, which increases the binding affinity to the host angiotensin converting enzyme 2 (ACE2) receptor, E484A, which is a site relevant for immune escape, and P681H, which enhances spike cleavage[43,44]. Due to the extensive divergence of Omicron from other circulating variants, further sampling of more related lineages would have been needed to resolve changes to the ones most likely to provide selective advantages only[45]. Taken together, the results demonstrate how the CoVerage allele dynamics analysis can identify emerging VOCs and their distinctive amino acid alterations based on the ecological dynamics of the spike protein allele in the viral epidemic, without their formal classification as novel lineages.

## Case study 1c: CoVerage predicts antigenic alterations for emerging Omicron variant

CoVerage provides an antigenic scoring analysis that identifies antigenically altered variants based on changes in the SARS-CoV-2 spike protein. In November 2021, Omicron (BA.1) was assigned an antigenic alteration score of 14.41 (standardized score of 7.81), more than six times that of Delta (B.1.617.2; 2.11 with a standardized score of −0.28) and one and a half times the antigenic score of Gamma (P.1; 5.23 for November 2021, no standardized score was assigned, as it did not meet the frequency threshold that month), in line with its more pronounced antigenic change[41]. The rapid spread of Omicron and its sublineages can also be observed from the global antigenic change maps for October, November,

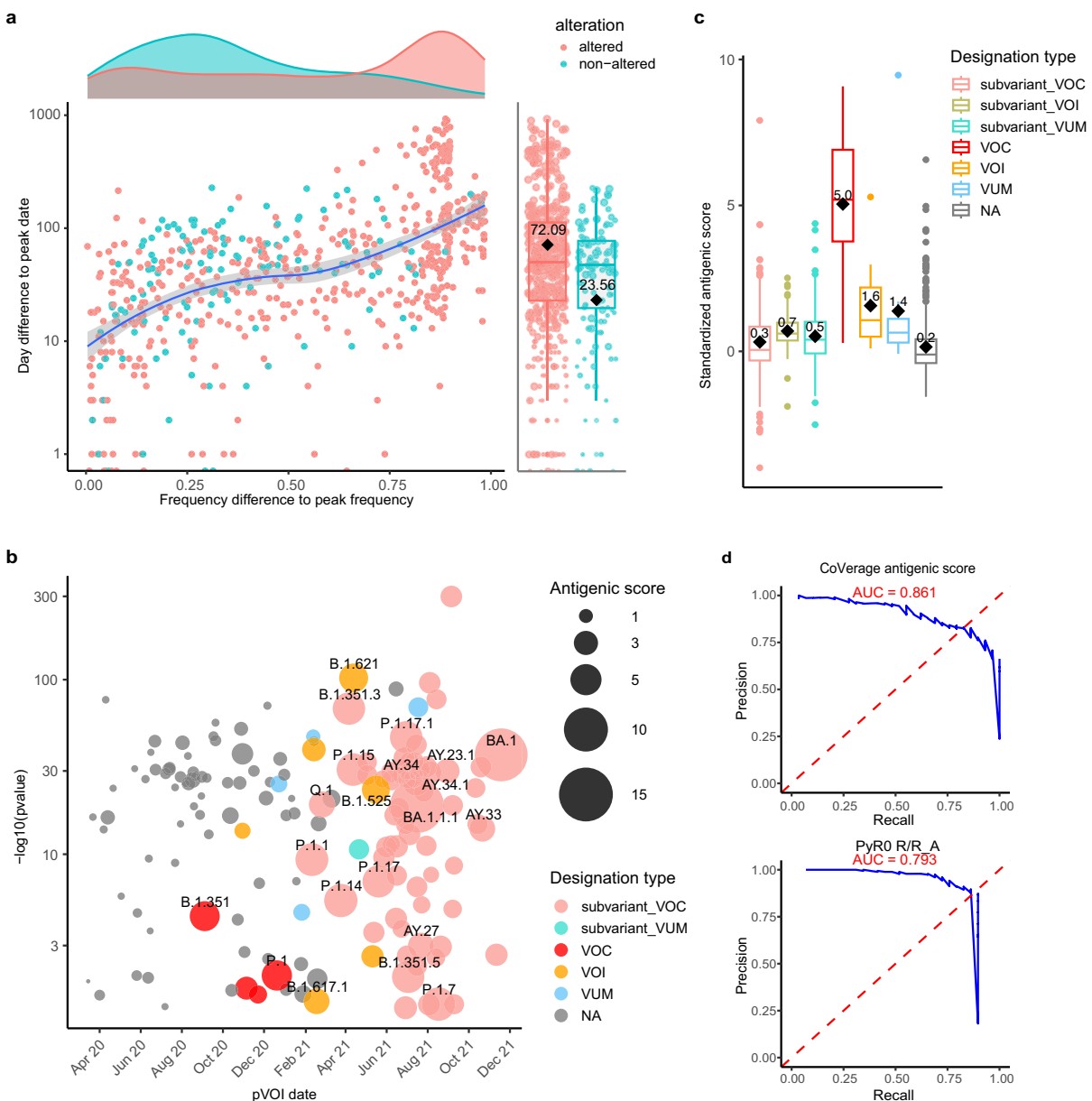

**Fig. 3 | Performance evaluation of CoVerage. a** The difference in days and frequency of pVOI identification of lineages relative to their peak frequencies in different countries. The dots in red indicate predicted antigenically altered pVOIs ($n = 674$), while the turquoise dots represent predicted non-altered pVOIs ($n = 193$). The pVOIs without standardized antigenic scores are excluded. A LOESS smoothed curve (blue line) summarizes the trend, with shaded confidence band showing the 95% confidence interval of the fit. Side histograms (top) and vertical strip plots (right) show the marginal distributions. **b** Antigenic scores of pVOIs identified using data before December 2021. Colors represent different WHO designations. The *x*-axis shows the month of the first identification, with "Apr. 20" indicating April 2020. The *y*-axis shows the $-\log_{10}$ of the adjusted p-value for pVOI detection, where $p$ values were calculated by a one-sided Fisher's exact test and corrected for multiple comparisons (see Methods for more details). The size of the points ($n = 153$) represents the antigenic score of the pVOI. **c** Standardized antigenic scores for variants and their subvariants for different WHO variant categories designations until June 2024. This analysis includes lineages categorized as subvariant_VOC ($n = 295$), subvariant_VOI ($n = 128$), subvariant_VUM (578), VOC ($n = 5$), 11 VOI ($n = 11$), VUM ($n = 12$), and non-designated ($n = 497$) lineages. **d** PRAUC of CoVerage antigenic score and PyR0 $R/R_A$ in predicting lineages as WHO-designated or non-WHO-designated. Data for both methods were collected before December 2021 (1683 lineages were analyzed), with the ground truth including WHO-designated lineages up to May 2022 to demonstrate early detection capabilities. In **b**, **c**, "NA" indicates lineages not designated by WHO. In **a** (side boxplot) and **c**, The horizontal bar within each box indicates the median, while the black diamond and the number above it represents the mean, the box bounds indicate the 25th (Q1) and 75th (Q3) percentiles; whiskers extend to the smallest and largest values within 1.5 times of the interquartile range (Q3–Q1); points beyond the whiskers are outliers. Source data are provided as a Source Data file.

and December 2021 (Fig. 5a, b, c, Methods), which reflect the antigenic alteration scores of lineages weighted by their frequencies per country. For South Africa, in November 2021 the country's antigenic score rose from 2.40 in October to 12.86 and was substantially greater than other countries' antigenic alteration scores at that time (standardized score of 6.939), reflecting Omicron's rapid spread through the country (Fig. 5b, d, e). Subsequently in December, the country antigenic scores increased globally, as Omicron and its sublineages rose in frequency, replacing previously circulating lineages (Fig. 5c). By

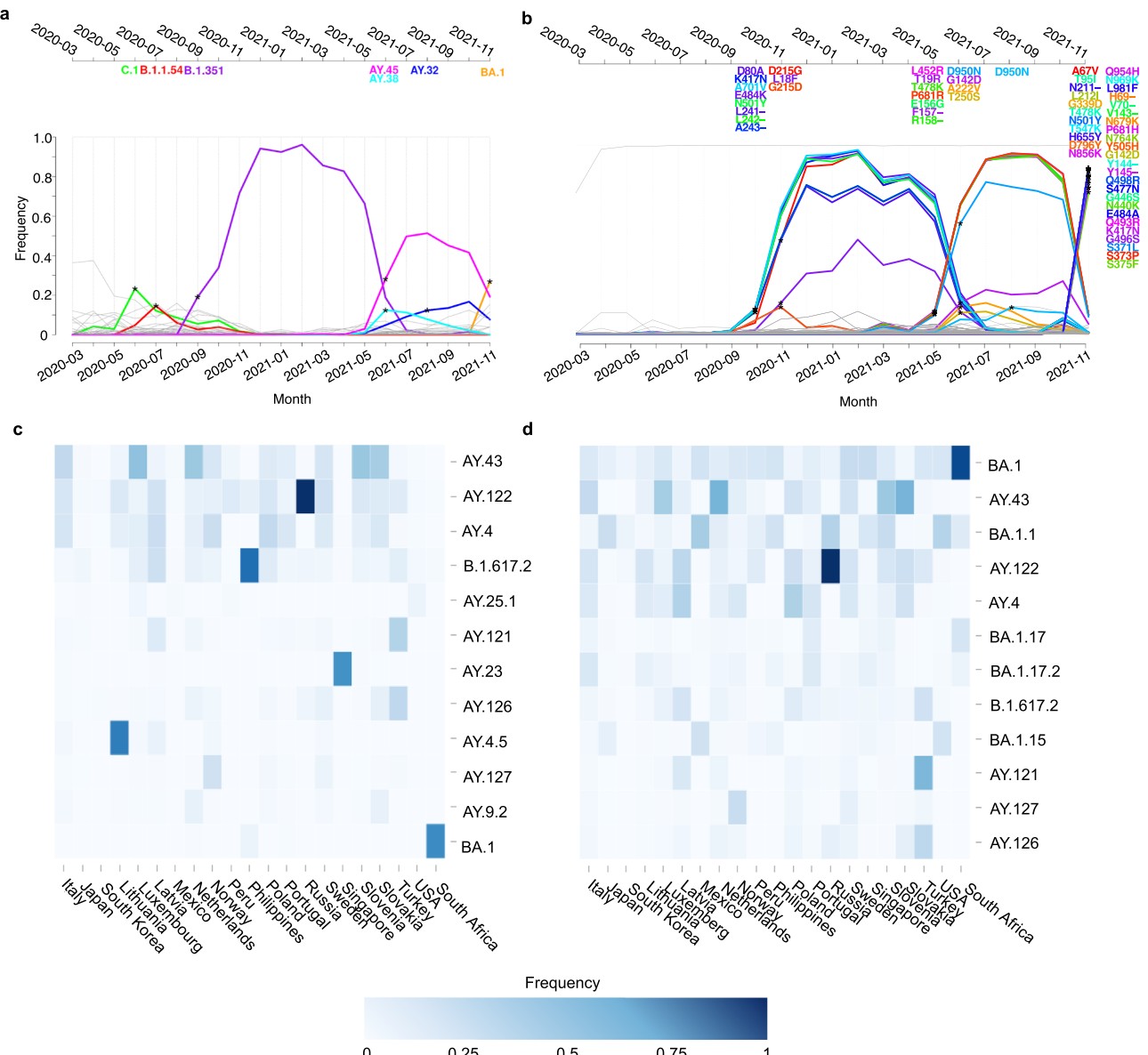

**Fig. 4 | Lineage and allele dynamics analyses through November 2021. a** Lineage dynamics plot for Pango lineages of SARS-CoV-2 genomes from South Africa submitted to GISAID by November 23rd, 2021 (total of 23,371 sequences). Colored lines represent identified pVOIs with their Pango lineage names given above. Asterisks indicate the month in which they were identified as being significantly on the rise and increasing in predominance above a predefined threshold of 0.1. All other lineages are shown in gray. **b** SD plot for South Africa on spike protein sequences available until the end of November 2021. Similar to the lineage dynamics plot, the asterisks represent when a spike protein allele with certain amino acid changes significantly rises in frequency, and the color of the curve corresponds to the associated amino acid change. Amino acid changes are specified at the top of the plot, based on the time point when they were identified as significant. The asterisks in November 2021 indicate the detection of the emerging Omicron lineage

(officially announced as a VOC on November 26th, 2021)[78], in May 2021 of Delta (Pango lineage B.1.617.2, detected with CoVerage first as a pVOI for India in March of 2021 and announced as a VOC on May 11th, 2021)[9], and Beta (B.1.351, announced as a VOC on December 29th in 2020) in September of 2020[79]. Asterisks in November 2020 indicate within-lineage variation of Beta, in June and August of 2021, of sublineages AY.45, AY.38, and AY.32 of the Delta variant, respectively. Relative frequencies for SARS-CoV-2 pVOIs identified in individual countries and most abundant worldwide. **c** November 2021 (calculated from a total of 742,970 sequences) and **d** December 2021 (calculated from a total of 907,496 sequences). The color scale ranges from dark blue, which indicates a frequency of one, to white, indicating a frequency of zero. The top 50 countries with the highest lineage frequencies are shown. Source data are provided as a Source Data file.

January 2022, BA.1.1.2, a sublineage of Omicron, rose to predominance in Japan, accounting for over 65% of the submitted sequences that month, significantly increasing its country antigenic alteration score (standardized score of 1.965; Fig. 5d). Soon after, BA.1.1, another sublineage of Omicron, also rose to predominance in Mexico with a frequency of greater than 60%, again significantly altering the country's antigenic score in comparison to other countries (standardized score of 2.197; Fig. 5d). Subsequently, sublineage BA.2.12.1 rose to predominance in the United

States in May 2022, also increasing its country antigenic score significantly (standardized score of 1.135). Each of these Omicron sublineages better evades neutralizing antibodies compared to BA.1[46], in line with their significant differences in assigned antigenic alteration scores. This pattern is also continued through the rest of 2022 and 2023 with the rise of further Omicron sublineages with significant antigenic alterations, e.g., for India and Singapore in September 2022 (standardized scores of 3.932 and 3.672, respectively), with the sublineages of BA.2 and BA.5

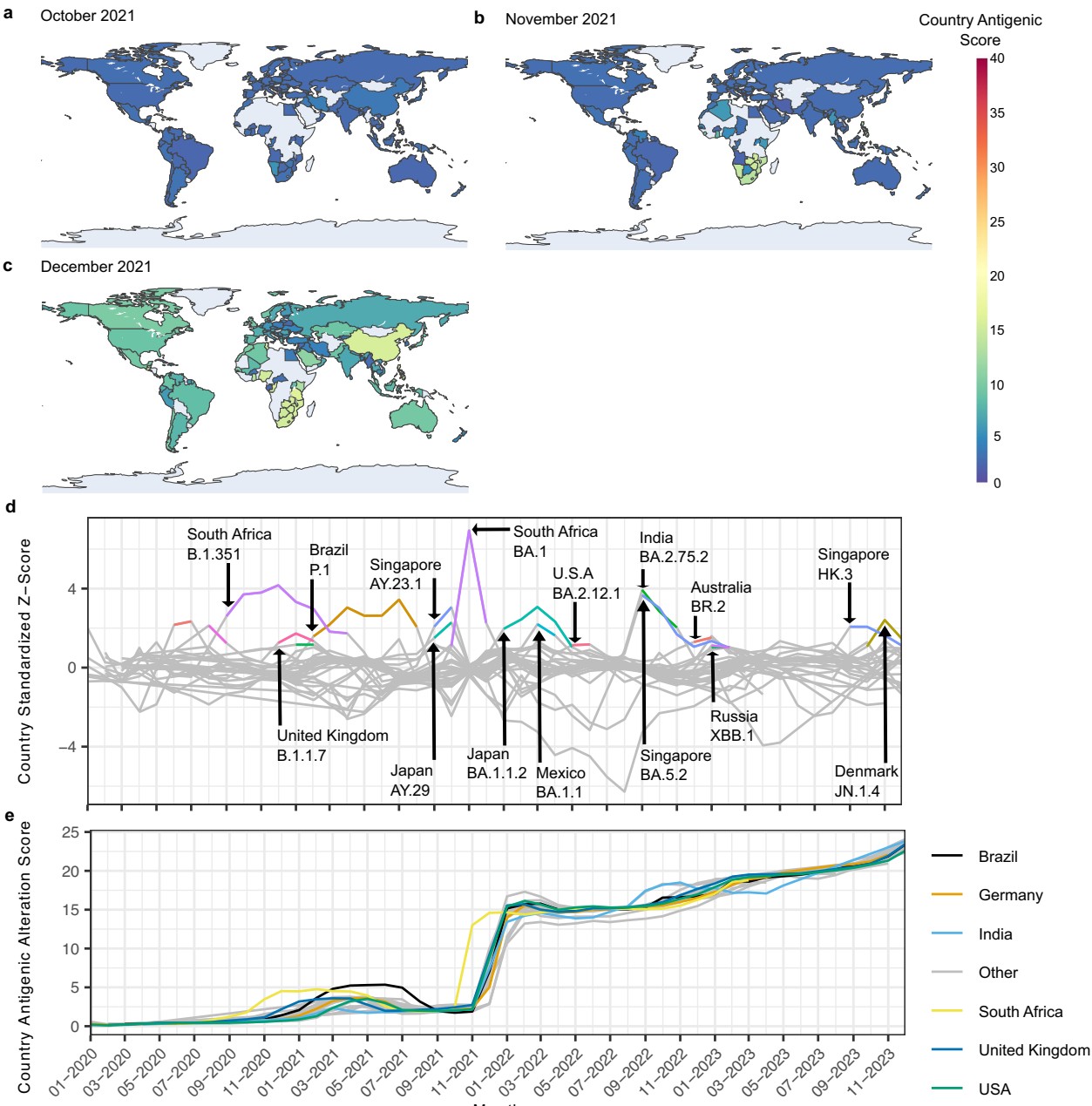

**Fig. 5 | Global spread of the antigenically altered Omicron lineage and its sublineages.** Global map depicting the assigned antigenic alteration score per country over the course of three months when Omicron became a predominant circulating VOC: **a** October 2021 (calculated from a total of 693,820 sequences), **b** November 2021 (calculated from a total of 858,797 sequences), and **c** December 2021 (calculated from a total of 1,119,014 sequences). A higher score (warmer colors) means that antigenically altered lineages circulate with higher frequency in that country. Dark blue represents an antigenic alteration score per country of 0, while red depicts an antigenic country antigenic alteration score of 40 (with arbitrary units). **d** Standardized antigenic scores per country for countries with either 1% of the monthly global sequences or a minimum of 500 sequences for at least 30 months. With this, there may be some missing months for a given country as they do not meet the required number of sequences. The standardized country

antigenic alteration score was calculated by subtracting the mean antigenic country score of the specific month from the country score and dividing by the standard deviation of the country score. Countries with a standardized score greater than or equal to 1 are shown in color for the month(s) they are above the threshold to show when a country's antigenic score is significantly greater than the other countries' scores for that month. **e** Plot of the country-wise antigenic alteration scores based on the circulating lineages in that country, their antigenic score, and their respective frequency for that month. Brazil, Germany, India, South Africa, the UK, and the USA are highlighted, while other countries with either 1% of the monthly global sequences or a minimum of 500 sequences for at least 30 months during the time period are represented in gray as "Other". Source data are provided as a Source Data file.

predominantly circulating. By October 2022, the XBB lineage and some of its sublineages, including XBB.1 and XBB.3, rose to predominance in India, which has an improved capacity for immune escape[47]. XBB.1 and its sublineages XBB.1.9.2 and XBB.1.5 also rose in frequency in Russia (with approximate frequencies of 18%, 8%,

and 4%, respectively) in January 2023, significantly altering the country antigenic score in comparison to other countries for that month. At the end of 2023, Denmark's country antigenic score significantly increased, with the predominant strain in October being HV.1, a XBB descendant with reduced antibody

neutralization[48], however, in November it was quickly overtaken by JN.1 (accounting for 14% of sequences that month) and its sublineage JN.1.4 (accounting for 16% of sequences that month).

Earlier VOCs, such as Alpha (B.1.1.7), Beta (B.1.351) and Gamma (P.1) also demonstrated this significant rise in country antigenic scores as they propagated across the globe (Fig. 5d, e), in line with their reduction in neutralization activity from sera of patients infected with a strain of the ancestral Pango lineage B[30]. Beta arose in South Africa as of September 2020 as one of the predominantly circulating lineages that month[49], with a significant difference in country antigenic score compared to other nations (standardized score of 2.602; Fig. 5d). Subsequently, the United Kingdoms' antigenic score significantly increased in December 2020 (standardized score of 1.252; Fig. 5d), in which Alpha was predominantly circulating, comprising approximately 45% of the sequences that month. Gamma (Pango lineage P.1) was identified in Brazil in December 2020 and rose to predominance over the next months[50], resulting in significant increases of the country antigenic score from 0.783 in October 2020 to 4.812 in March 2021 (Fig. 5d, e). By utilizing the country antigenic scores as visualized in the global antigenic change maps and standardized country-wise z-plots, novel SARS-CoV-2 lineages with a higher capacity for immune escape can be identified as pVOIs contributing to increasing country-wise antigenic scores as they rise to predominance.

### Case study 2: screening with CoVerage for antigenically altered pVOIs in 2023

From available data until 31 March 2023 (Supplementary Tables 8–10), 48 lineages were identified as pVOIs for January, 46 lineages in February, and 37 lineages in March with CoVerage. Most of these are sublineages of BQ.1 and XBB, denoted as BQ.1* and XBB* as per WHO specification[42]. Using CoVerage, the BQ.1 lineage, which was first identified in Nigeria[51], was detected as a pVOI also in Nigeria in August 2022 (Fig. 6a) and designated as a Variant of Interest by the WHO on September 21st, 2022[52]. In the final quarter of 2022 and in the beginning of 2023, BQ.1* was one of the major circulating lineages globally, with BQ.1 and the sublineage BQ.1.1 together covering over 50 percent of submitted sequences in January 2023[33]. Of the XBB sublineages, XBB.1.16 was identified by CoVerage as a pVOI in India in February 2023, two months prior to its VOI designation on April 17, 2023 (Fig. 6b)[42]. XBB.1.5 was identified as significantly antigenically altered in November 2022, with an alteration score of 19.34 (standardized score of 1.78)[53], prior to its VOI and pVOI designations in January 2023 (Fig. 6c). BQ.1 and sublineages were quickly overtaken by the XBB.1.5 lineage, which in March 2023 represented over 50% of global sequences[54]. In the antigenic alteration maps, the change from BQ.1.1 to XBB.1.5 as one of the major circulating lineages is evident from January, February, and March 2023 (Fig. 6d, e, f). Finally, the WHO recommended an update of COVID-19 vaccine formulations to include the XBB.1.5 lineage in May 2023[55].

The increased antigenic alteration scores of the Omicron BQ* and XBB* lineages align with their demonstrated capacity for immune evasion and with their reduced neutralization by select mAbs[33]. Key amino acid changes on the spike protein facilitate immune escape, most notably F486P on the receptor binding domain of XBB.1.5[56] (Supplementary Fig. 4), which also increases infectivity by improving the binding of the spike protein to host ACE2 receptor cells[57]. Furthermore, the N460K mutation of BQ.1.1 reduces the neutralizing activity of NTD-SD2 and class I mAbs, while the K444T and R346T changes in BQ.1.1 may also impair the potency of class III mAbs[33].

Towards the end of 2023, the JN.1 lineage, a BA.2.86 descendant, was designated a Variant of Interest by the WHO[58], two months after its demarcation as a pVOI in Portugal via the lineage dynamics analysis and one month after its contribution to the rising country-wide standardized score of Denmark in the country-wise standardized score analysis (Fig. 5d). A key amino acid change is L455S[59], which

was identified e.g., via the allele dynamics analysis in December 2023 in Germany, along with the L157S, N450D, L452W, and N481K changes. Both the N450D and the L452W amino acid changes on the receptor binding domain (RBD) of the spike protein improve viral evasion from multiple mAbs[60]. In experiments, JN.1 showed more extensive resistance to RBD class 1, 2, and 3 antibodies as well as to monovalent XBB.1.5 vaccine sera compared to BA.2.86, which is also reflected in its increased antigenic score for December 2023 of 23.65 (standardized score of 0.59) versus 21.33 (standardized score of −0.48), respectively[59,61]. In late April, the WHO advised that future formulations of the COVID-19 vaccines should focus on JN.1[16]. Subsequently, due to the continued evolution of SARS-CoV-2, the recommendation for the preferred strain was changed to the KP.2 sublineage[62].

## Discussion

Due to the ongoing and rapid genetic and antigenic evolution of circulating SARS-CoV-2 viruses, detecting emerging Variants of Concern that are on the rise to predominance as early as possible is crucial for public health decision-making. This included nonpharmaceutical interventions in the early stages of the pandemic and, more recently, updating the vaccine composition to ensure continued effectiveness[16,55]. Here, we describe CoVerage, a genomic surveillance platform that continuously monitors globally provided SARS-CoV-2 genomics data to identify and characterize potential Variants of Interest from the circulating lineage diversity. CoVerage implements three types of innovative methods for this purpose: (a) a method for the de novo detection of potential Variants of Interest that may spread more efficiently than others, (b) a method to identify amino acid changes in the major surface spike protein that may confer a selective advantage, and (c) a method for scoring the degree of antigenic alteration of individual sequences, lineages and the circulating viral diversity per country. To ensure maximal relevance for viral surveillance, CoVerage provides up-to-date predictions of current pVOIs and their antigenic alterations for all countries around the globe with sufficient data available once a week.

In a systematic assessment of the combination of CoVerage pVOI predictions and its antigenic alteration assessments, CoVerage accurately identified 88% of the VOIs and VOCs designated by the WHO since the establishment of SARS-CoV-2 in the human population, on average, more than two months before their official WHO designation. When including the VUMs, the pVOI predictions achieved a precision of 79% and recall of 72%, and CoVerage antigenic alterations scores in PRAUC analyses also demonstrated high predictive value. The pVOIs and antigenically altered variants identified by CoVerage thus comprise a highly informative set of variants with a potential selective advantage for identifying the VOIs, VOCs, and VUMs defined by the WHO. While the identification of amino acid changes throughout the spike protein that may have selective advantage can further inform antibody design and help understand the molecular basis of adaptive evolution[63]. CoVerage predictions are fully reproducible as they are derived from a defined set of input data, with a fully deterministic, statistical assessment of globally available data, to support and facilitate further expert assessments. Notably, CoVerage's detection depends on the extent and quality of ongoing viral genomic surveillance programs for individual countries, as the analysis is done in a country-wise manner and may also be affected by population genetic effects[64], such as population bottlenecks, when case numbers are low, or travel restrictions between countries are in place, such as in early phases of the pandemic. In terms of data, CoVerage draws on the international GISAID data resource and combines it with other repositories where data is available in advance, to decrease the time to detect new, relevant variants[65]. Detection may be affected if genomic surveillance would be decreased further in the future with the virus becoming endemic.

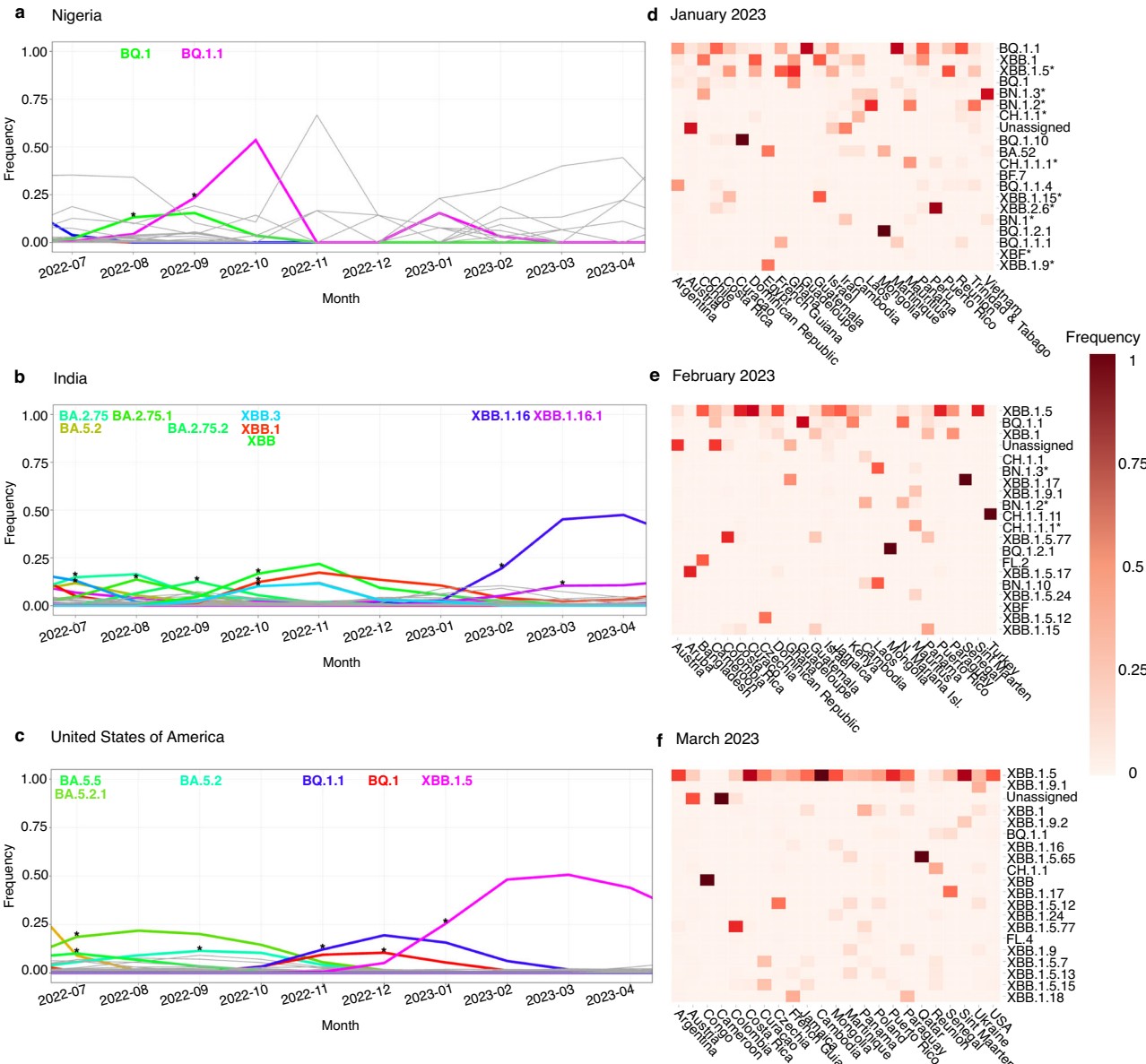

**Fig. 6 | Lineage dynamics plots and heatmaps of antigenically altered lineages through early 2023.** Country-wise lineage dynamics plots for **a** Nigeria, including results from July 2022 to April 2023, **b** India, including results from July 2022 to April 2023, and **c** the United States, including results from July 2022 to April 2023. **a** The BQ.1 lineage is shown in green and the BQ.1.1 lineage in pink. The asterisks represent a significant rise in frequency, which is seen in August 2022 for BQ.1. **b** The XBB lineage is shown in bright green (October 2022) and the XBB.1.16 lineage is shown in dark blue, XBB.1 in red, XBB.3 in light blue, and BA lineages are shown in shades of green (BA.2.75 and BA.5.2 in July 2022, BA.2.75.1 in August 2022, BA.2.72.3 in September 2022). **c** BA lineages are shown in shades of green and light blue

(BA.5.5 and BA.5.2.1 in July 2022, BA.5.2 in September 2022), BQ.1.1 in purple, BQ.1 in red, and XBB.1.5 in pink. The asterisk represents a significant rise in frequency for that lineage. Heatmaps with antigenically altered lineages and their relative frequencies per country in **d** January 2023 (total of 246,312 sequences) **e** February 2023 (total of 184,910 sequences), and **f** March 2023 (total of 176,324 sequences). The lineages are ordered from highest to lowest frequency, and the dark red color indicates a frequency of 1, while the lighter color red represents a lower frequency closer to 0. Significantly antigenically altered lineages (with a standardized score greater than or equal to 1) are denoted with an asterisk. Source data are provided as a Source Data file.

CoVerage identifies antigenically altered lineages with a method that assesses all changes in the spike protein, which is the key immunogen and protein for the binding of SARS-CoV-2 to host receptor cells, and as such is a target for vaccines and antibody therapy[26]. By weighing amino acid changes occurring across the entire spike protein, the method accounts for any changes in key antigenic sites as the virus continues to evolve. An initial list of antigenic sites across the spike protein that we compiled with literature through 30 June 2021, was not comprehensive enough to cover the continued antigenic evolution of the virus, as it mostly includes sites identified from the Alpha, Beta and Delta variants and

lacks key antigenic sites such as 59, 460, 456, or 505 that arose in later Omicron lineages such as XBB, EG.5 and XEC[33–36]. We show that this method in combination with pVOI predictions identifies priority variants and their sublineages, which were defined by the WHO as of public health relevance, such as VOCs, VOIs, and VUMs. In evaluating antigenic alteration predictions, we found that predicted antigenic alteration scores are reflective of how well the evolutionary changes affect the neutralization achieved by antibodies. Studying antigenic scores, acquired mutations, and their positioning on the 3D structure might reveal further insights into the mechanistic basis of varying viral variant neutralization. Additionally, by using the country

antigenic scores lineages with a higher capacity for immune escape could be identified on a country-wise basis as they rose to predominance, after their initial detection as pVOIs, including early antigenically altered VOCs such as Alpha, Beta, and Gamma, as well as Omicron and its subsequent sublineages such as XBB.1 and JN.1.

In several case studies, we show how CoVerage detected relevant VOCs as pVOIs along with their antigenic alterations in a timely manner and tracked their ongoing, global spread. In November 2021, Omicron was identified as a pVOI and assigned a substantially increased antigenic score compared to previously circulating lineages. Consistent with the rapid designation of Omicron as a VOC by the WHO, which, in addition to sequencing information, also considers epidemiological evidence, CoVerage identified this pVOI from the submitted data just three days before. In early 2023, the majority of pVOIs identified by CoVerage were sublineages of BQ.1 and XBB, both of which have demonstrated improved immune escape due to key mutations throughout the spike protein[33], which were also found in the spike protein allele dynamics plots of individual countries. By May of 2023 the WHO recommended updating vaccines to match XBB.1.5, which had been identified as a significantly antigenically altered lineage in November 2022 by CoVerage, and subsequently, updated vaccine recommendations to match the circulating JN.1 in April 2024, which had also been identified as an antigenically altered pVOI in October 2023[16,55]. Altogether, this shows that the combination of analyses provided by CoVerage facilitates the timely detection and characterization of relevant SARS-CoV-2 lineages for public health concerns.

Taken together, CoVerage is a unique web-based resource to identify pVOIs in a timely manner, along with suggesting their degree of antigenic alteration, and alleles of the major surface protein with specific amino acid changes that may provide a selective advantage. There are other web-based resources to track SARS-CoV-2 variants, among other viruses, and their viral fitness and evolution. NextStrain, for instance, not only established a viral lineage nomenclature based on phylogenetic principles for SARS-CoV-2, but also continuously assesses logistic growth rates, immune escape in comparison to BA.2, and mutational fitness per lineage[66]. Similarly, PyR0 is a hierarchical Bayesian multinomial logistic regression model that detects lineages increasing in prevalence as well as identifies mutations relative to lineage fitness[39], and Episcore predicts which existing amino acid mutations might contribute to future SARS-CoV-2 VOC's[67], though neither is run continuously nor available as a web-based platform. Other web-based platforms, such as CoVariants, provide an overview of variant frequencies and shared amino acid mutations[68], and CovidCG tracks viral mutations, lineages, and clades in different countries over time[69]. CoVRadar focuses on mutation frequency by location and mutation distribution among sequences for the molecular surveillance of the SARS-CoV-2 spike protein[70]. Outbreak.info also provides information about lineages and amino acid changes while also reporting prevalence of variants, their geographical distributions, and comparisons of changes between lineages[54]. Both EVEscape and SpikePro, alternative platforms that score variants on potential immune escape and fitness, correlated slightly less than the antigenic alteration predictions with antigenic cartography distances, and ultimately, EVEscape does not identify variants with potential selective advantage, and SpikePro does not continuously score circulating lineages as CoVerage does[37,38]. Only CoVerage identifies variants with a potential selective advantage using lineage frequency dynamics in combination with predictions of lineage antigenic alterations. It also links to alternative web-based resources for additional information on these selected lineages, providing a comprehensive resource for lineage surveillance. Each of the different analyses provided on CoVerage offers its own benefits when used individually and more so in combination with one another.

CoVerage provides a continuously updated resource for the in silico detection and characterization of potential VOIs, VOCs, and VUMs from SARS-CoV-2 genomic surveillance data, to support researchers and public health officials in assessing and interpreting such data for public health decision making and vaccine updates. In the future, the framework will be extended to provide relevant surveillance for other rapidly evolving viruses, such as seasonal influenza viruses, to track the geographic spread of lineages and to integrate further data types, such as metagenomic sequences of wastewater samples[71].

## Methods

### Lineage dynamics by region

In the sublineage corrected analysis, the lineage frequencies are corrected by including isolates belonging to sublineages in the count, and subsequently processed as before. For example, three sublineages BA.1.2, BA.1.3, and BA.1.4 will be summarized into BA.1 and then analyzed with one-sided Fisher's exact test (R fisher.test, alternative = "greater"). Multiple testing was corrected using Benjamini–Hochberg procedure[22]. This approach leads to a better resolution of the lineage branches, likely associated with more rapid spread, and lineages with a selective advantage split into several sublineages can be better detected. Here, Pango lineages were used as nomenclature as they define an epidemiologically relevant phylogenetic cluster in which new lineages are only designated if the lineage has high coverage and contains a sufficient number of sequences[21].

Lastly, a sliding window approach was implemented to achieve a more sensitive detection than the monthly analysis. For this analysis, sequences are sorted by date, and frequencies in the $w$ sequences in the current window are compared to the $w$ previous ones using one-sided Fisher's exact test (R fisher.test, alternative = "greater"). The window is moved over the data using a step size $s$. Significance estimates are corrected for multiple testing through correction of $p$ values of the multiple tests with Benjamini–Yekutieli procedure[72]. For the analysis on countries, $w = 1000$ and $s = 100$ are used, and for the analysis on more granular German state level $w = 200$ and $s = 10$ are chosen. Additionally, we require a significant (FDR < 0.05) increase for a certain window and a frequency threshold of 0.1 to report results. The reported date, or the date of the significant increase, is the date of the last sequence in the current window.

### Spike protein allele dynamics

To identify amino acid changes in the spike protein that may provide a lineage with a selective advantage, the sweep dynamic (SD) plot method[17,20] is used on spike protein sequences extracted from viral genome sequences data downloaded from GISAID and Zenodo. To execute this methodology on the large SARS-CoV-2 sequence collection, 500 sequences per month per country are subsampled randomly and downloaded with GISAIDR[73], and subsequently, identical sequences per time period are clustered using CD-HIT[74]. For German state-wise analysis, we randomly downloaded 2000 sequences per month for the entire country and divided them by state. Next, a multiple sequence alignment is generated using MAFFT[75], with the spike sequence of Wuhan/IPBCAMS-WH-01/2019 as a reference. Phylogenetic trees are inferred for each country using fasttree[76], and the Sankoff algorithm is applied for ancestral character state reconstruction, and as in Steinbrueck et al. we use the same cost for all amino acid changes and indels[20]. We previously showed that in ancestral sequence reconstruction for the major antigen of human influenza A (H3N2) viruses, both approaches produce very similar results, due to the very small evolutionary time scales being considered[20]. The use of ancestral sequences reconstruction allows for the identification of amino acid change events in the evolutionary history of SARS-CoV-2 lineages, which defines lineage-alleles and their frequencies within a specific

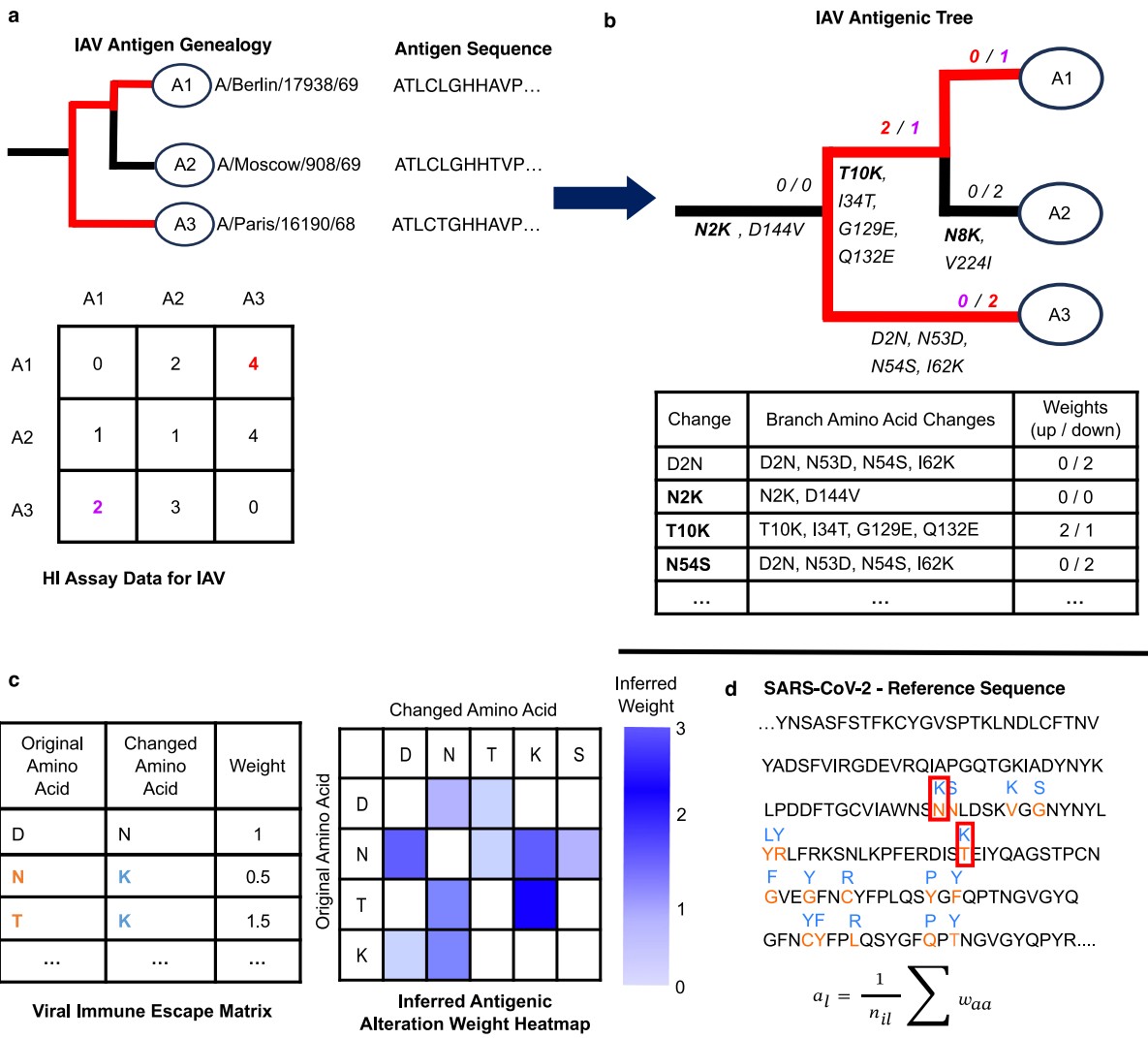

**Fig. 7 | Schematic outline of the technique to predict the antigenic alteration of SARS-CoV-2 spike protein sequences using a IAV-derived viral immune escape matrix. a** The inputs for antigenic tree inference are Hemagglutination Inhibition Assay (HIA) data table for IAV and a genealogy inferred from isolate sequences of the major viral surface protein, and the corresponding amino acid sequences[63] **b** Antigenic tree with antigenic branch weights inferred and amino acid changes in the viral antigenic reconstructed for the evolutionary branches. Values such as "0/2" indicate antigenic up and down branch weights, reflecting the asymmetric nature of hemagglutination inhibition data, where antigenic distance from a viral antigen strain *a* to antiserum raised against another strain *b* can be different from the distance of an antiserum raised against antigen *a* to an antigen *b*, respectively. The red color traced through the IAV Antigenic Tree indicates an example path between the A1 and A3 variants, for which the sum of edge weights corresponds to the red values for A1–A3 and purple values for A3–A1 in the HIA table in (**a**). Several changes in bold are exemplary shown in the IAV Antigenic Tree table, which represents for every position-specific amino acid mapped to a tree branch the antigenic branch weights. The up/down weights inferred in this tree are then compiled into **c** position-agnostic viral immune evasion antigenic alteration matrices for pairs of amino acids, shown here with the example of averaging over inferred up and down weights. **d** These weights are then used to score amino acid changes across the spike protein of SARS-CoV-2 to calculate sequence-specific antigenic alteration scores, which are then combined in lineage and country-wise alteration scores.

time period as the ratio of the number of isolates in the subtree of the allele relative to the number of all isolates within the designated period[20]. Despite being more complex, this gives a deeper evolutionary understanding of the dynamics of individual lineages and their associated amino acid changes and thus determines whether associated changes increase significantly over time, potentially providing selective advantage[20]. From the inferred phylogenetic tree, each branch associated with a non-empty set of substitutions is set to represent an individual allele, and the frequency of the alleles within a given time period is thus defined as the ratio of the number of isolates in the subtree of the allele relative to the number of all isolates with the given period[20]. The frequencies of the alleles between the different time periods are then compared using a one-sided Fisher's Exact Test to identify alleles significantly on the rise in frequency[20].

## In silico assessment of antigenic lineage alterations

We quantify the degree of antigenic divergence of a viral isolate relative to another one using an antigenic scoring matrix, which includes antigenic alteration estimates for pairs of amino acid changes derived from a previously inferred antigenic tree for Influenza A viruses (IAV)[63]. These were established by inferring antigenic weights, i.e., antigenic branch lengths, from hemagglutination inhibition (HI) assay data onto a maximum likelihood phylogenetic tree and matching these to branch-specific sets of amino acid changes inferred by ancestral character state reconstruction for the major viral surface protein for human influenza A (H3N2) viruses, combining genetic with evolutionary and serological data (Fig. 7a, b). The inference of the antigenic tree is based on a least-squares optimization (LSO) procedure, which fits dissimilarities derived from HI assay data onto an Influenza A virus

(IAV) evolutionary tree (Fig. 7a)[63]. The branch length is then representative of the antigenic difference between nodes in the genealogy, which, after reconstructing amino acid changes via ancestral character state reconstruction and mapping these onto the tree branches throughout the tree, indicates the potential impact of the branch-associated amino acid changes on the immune escape of the virus. As HI assay data is asymmetric, the antigenic distance from a viral antigen strain $a$ to antiserum raised against another strain $b$ can be different from the distance of an antiserum raised against antigen $a$ to an antigen $b$, respectively. The antigenic tree uses both up and down weights for each branch, where the up weights are representative of the antigenic distance from isolates below the branch to all other isolates, and the down weights are representative of the distances from isolates outside the subtree to isolates below the branch[63]. It is this combination of up/down weights and amino acid changes throughout the inferred IAV antigenic tree (Fig. 7b) that is used to create a viral immune escape matrix (Fig. 7c), which includes one antigenic alteration weight for each amino acid change. These weights can then be applied to amino acid changes occurring across the SARS-CoV-2 spike protein and quantify the antigenic alteration of SARS-CoV-2 strains relative to the originally described SARS-CoV-2 isolate (GenBank MN908947.3) or reference sequence (Fig. 7d).

From the inferred IAV antigenic tree, several antigenic alteration matrices specifying weights for all pairs with amino acid changes with evidence in the antigenic tree were defined. The weights serve as a measure for the potential impact that the amino acid substitution has on the antigenicity of the variant. M_ant_alt_0 includes weights averaging over the respective up or down weights seen for a particular change across all positions, such as the change N_K, assigned a weight of 0.5, averaging over $(0+0+0+2)/4$ (Fig. 7c). Matrix M_ant_alt_1 includes weights from M_ant_Alt_0 averaged across change and reverse change. The matrix M_ant_alt_2 includes a total of forty-one weights, and matrix M_ant_alt_2 includes antigenic weights only for changes occurring at least three times in the antigenic tree. All unspecified pairs of changes in the antigenic matrices include missing values and are not counted (https://github.com/hzi-bifo/Corona_Variant_Scoring/tree/main/reference).

We then explored whether antigenic weights for pairs of amino acid changes derived from data for the antigenic evolution of human influenza A (H3N2) viruses are predictive for the degree of antigenic evolution of human SARS-CoV-2 viruses. Specifically, the antigenic alteration score $a_l$ for a lineage $l$ is calculated to predict the antigenic alterations of each SARS-CoV-2 spike protein sequence thereof relative to the reference strain. To this end, antigenic weights from the viral immune escape matrix are applied to the amino acid changes of a lineage throughout the spike protein relative to reference strain. The weights are subsequently summed per isolate and averaged across all isolates of a given circulating Pango lineage to calculate the final antigenic score, $a_l$. To select lineages considered significantly altered antigenically for a given month, a z-score standardization was applied to each circulating Pango lineage for the given month with a frequency of 0.001 or greater. To calculate the standardized score, the population mean, or the mean of the lineages circulating that month above the frequency threshold, was subtracted from the individual lineage score and then divided by the standard deviation of those lineages circulating above the frequency threshold. Circulating lineages with an assigned standardized score of one or greater were considered to be significantly antigenically altered.

For scoring circulating SARS-CoV-2 lineages, we used the metadata provided by GISAID and filtered isolates by the most recent complete month (i.e., the previous month) based on the isolate collection date. Sequences under review were removed. Among other information, the metadata contains amino acid substitutions or deletions occurring in each isolate as well as their Pango lineage, which would be used to assign an antigenic weight to each change occurring at the previously defined sites. Once assigned to each change, the individual amino acid weights were then summed per isolate and then averaged across each Pango lineage occurring in the selected month to assign one final antigenic score to each circulating Pango lineage that month, such that:

$$a_l = \frac{1}{n_{il}} \sum w_{aa} \tag{1}$$

where $a_l$ is the antigenic score for the Pango lineage $l$ and $n_{il}$ represents the number of isolate sequences per Pango lineage in the given month and $w_{aa}$ represents the assigned weight for an amino acid change.

In addition to the individual lineage scores, a country-wide antigenic alteration score, $a_c$, is calculated by summing over all antigenic lineage scores, weighted by their frequencies for a particular period in time in a country $c$:

$$a_c = \sum \left( a_l \times \frac{n_{ilc}}{n_{ic}} \right) \tag{2}$$

Here, $n_{ilc}$ is the number of isolates of each Pango lineage for country $c$ and $n_{ic}$ is the total number of isolate sequences per country.

## Creation of datasets for antigenic lineage alterations method comparison and validation

Antigenic cartography distance of selected variants of concern (Alpha, Beta, Delta, Gamma, Lambda, Mu, Epsilon, BA.1, and other Omicron sublineages)[27] as well as mean mFRN values[11] were used as a validation of calculated antigenic alteration scores. The geometric mean fold reduction in neutralization (mFRN) values of monoclonal antibodies collected in Cox et al. were averaged across all Class 1, 2, and 3 mAbs and mAb cocktails presented in Fig. 1b of the review, to get the mean mFRN values. The mFRN values were calculated in Cox et al. by measuring the concentration of mAb required to prevent the infection of cells by a virus carrying the mutated SARS-CoV-2 spike protein in comparison to the wild-type control sequences[11]. Therefore, a higher value represents a higher concentration of mAb to prevent infection, i.e., a variant with a higher potential for immune escape.

The antigenic cartography distances used as a comparison were derived from a merged antigenic map from human and hamster sera neutralization datasets available in Table S1 of the Wang et al. preprint[27]. These distances are representative of the antigenic relationships of SARS-CoV-2 lineages as they use antibody titers to visualize antigenic differences among strains[27].

Correlations between the antigenic scores and both the mFRN values and antigenic cartography distances were calculated using Spearman's correlation coefficient[77], and the correlations between the baseline methodology and the scoring method using weights at all amino acid sites on the spike protein were compared by normalizing the scores and the ground truth antigenic distances using min-max normalization and then comparing the differences between the normalized ground truth and antigenic scores from the different methods (method 2, which applied antigenic weights at all amino acid sites on the spike protein and the baseline method which counted all changes occurring at the spike protein) using a one sided Wilcoxon signed-rank test (R cor.test, alternative = "less", exact = FALSE, paired = TRUE; Supplementary Table 3)[31].

## Selecting lineages with significant antigenic variation

To select lineages with significant antigenic variation in comparison to other circulating lineages, each circulating lineage was assigned a standardized score based on its monthly antigenic score using z-standardization. To calculate the standardized score, the population

mean, or the mean of the lineages circulating that month above the frequency threshold, was subtracted from the individual lineage score and then divided by the standard deviation of those lineages circulating above the frequency threshold. Lineages with a standardized score of greater than or equal to 1 were ultimately selected as antigenically distinct from the other circulating lineages, or within the top 15% of circulating lineages. A standardized score of 1 was selected as a threshold, as on average WHO-defined variants of concern (VOCs) had an average standardized score of 1.2 (Supplementary Fig. 5).

### Reporting summary
Further information on research design is available in the Nature Portfolio Reporting Summary linked to this article.

## Data availability
Supporting data, including computationally derived data generated in this study, have been deposited in the Zenodo database under accession code 10171227. The processed data used for figure generation in this study are provided in the Supplementary Information/Source Data file. The genomic data used in this study were from GISAID and had been previously reported. The GISAID accession IDs used in this study are available in the GitHub repositories: https://github.com/hzi-bifo/corona_lineage_dynamics. https://github.com/hzi-bifo/Corona_Variant_Scoring. Source data are provided with this paper.

## Code availability
All code used in this study has been archived on Zenodo and is publicly available: Lineage Dynamics: https://doi.org/10.5281/zenodo.15311135 (GitHub: https://github.com/hzi-bifo/corona_lineage_dynamics). Protein Dynamics: https://doi.org/10.5281/zenodo.15311152 (GitHub: https://github.com/hzi-bifo/corona_protein_dynamics). Variant Scoring: https://doi.org/10.5281/zenodo.15322216 (GitHub: https://github.com/hzi-bifo/Corona_Variant_Scoring).

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

## Acknowledgements

We gratefully acknowledge all data contributors, i.e., the Authors and their Originating laboratories responsible for obtaining the specimens, and their Submitting laboratories for generating the genetic sequence and metadata and sharing via the GISAID Initiative, on which this research is based. Z.-L. Deng gratefully acknowledges funding from the German Centre for Infection Research project TI 12.002. Special thanks to Dr. Fernando Meyer and Hesham Almessady for testing available source codes.

## Author contributions

K.N., Z-L.D., G.R., S.G., M.H.F.A., M.H., and S.R. developed the methods and wrote the code. K.N., S.R., and A.C.M. analysed and interpreted the data, K.N., S.R., and A.C.M. wrote the paper, and A.C.M. designed the research study. M.H. and F.K. suggested data visualizations and commented on the manuscript. Z-L.D., K.N., G.R., M.H.F.A., and S.G. maintain the web service. All authors read and approved the manuscript.

## Funding

## Competing interests

The authors declare no competing interests.
