## [Transparent Peer Review file · Nature Communications]

In silico genomic surveillance by CoVerage predicts and characterizes SARS-CoV-2 Variants of Interest

Corresponding Author: Professor Alice McHardy

Version 0:

Reviewer comments:

Reviewer #1

(Remarks to the Author)

The paper by Norwood et al. introduces CoVerage, a web-based platform designed to monitor the spread of SARS-CoV-2 lineages worldwide and identify mutations in the spike protein that may have antigenic significance. While an automated framework for identifying variants with potential public health implications is beneficial, I am uncertain about the added value of this approach over existing methodologies:

1- Comparison with existing methods:

Several existing methods already robustly estimate the growth rate of lineages over time, assess the fitness advantage of mutations, and explore their implications for the antigenic evolution of the virus (e.g., <https://www.science.org/doi/full/10.1126/science.abm1208> and <https://www.science.org/doi/10.1126/scitranslmed.abk3445>). Although CoVerage was compared to some of the online dashboards like NextStrain, CoVariants, and CovidCG, the functionalities of at least some of these platforms were not accurately described in the paper. For example, NextStrain does more than just establish viral lineage nomenclature; it also has features that allow users to assess logistic growth rates and mutational fitness per lineage, features not adequately acknowledged in the comparison.

2- Antigenic alteration analysis:

A major feature of CoVerage that the authors emphasized was its ability to evaluate antigenic lineage alterations over time. However, I was surprised to see that the methodology for this analysis is still 'in preparation', and the specifics of the scoring system are unclear. The examples provided, such as the identification of the S:N501Y mutation in various variants, do not convincingly demonstrate the method's effectiveness, especially given the known saltatory evolution of SARS-CoV-2 and the emergence of highly divergent variants like the Omicron BA.2 and BA.5 lineages will usually lead to noticeable changes in the virus phenotype (see <https://www.nature.com/articles/s41579-023-00878-2> and <https://www.nature.com/articles/s41579-022-00841-7>).

4- Timing of variant identification by CoVerage vs. WHO announcements:

The emphasis by the authors on the timing of the identification of variants by CoVerage compared to WHO announcements can be misleading. WHO decisions are based on sequence data and epidemiological evidence available at the time, often much delayed from collection to submission to GISAID or other online platforms. Stating that CoVerage identified Omicron before its official designation by WHO overlooks the swift response by Southern African scientists that led to its rapid identification as a VOC. In fact, Omicron was indeed a good example of a swift announcement from WHO about its designation as a VOC thanks to the work of the Southern African scientists. On a separate note, the term 'potential variant of interest (pVOI)' used in the paper could contribute to confusion, as it lacks a clear definition and differs from established WHO terminology. A term like 'variant under monitoring' might align better with existing nomenclature without adding ambiguity (see <https://www.who.int/publications/m/item/updated-working-definitions-and-primary-actions-for--sars-cov-2-variants>).

5- Evaluation of method effectiveness:

It seems to me that the paper selectively presents instances where CoVerage successfully identified lineages and mutations with clear growth advantages. And there is a clear lack of a comprehensive evaluation of the method's sensitivity and specificity. The frequency of false positives, where variants are incorrectly labeled as pVOIs, is not discussed, which is crucial for assessing the reliability of the platform in early and reliable detection of variants that need to be monitored.

(Remarks on code availability)

Reviewer #2

(Remarks to the Author)

"In silico genomic surveillance by CoVerage predicts and characterizes SARS-CoV-2 Variants of Interest" by K Noorwood, ZL Deng, S Reimering, G Robertson, MH Foroughmand-Araabi, S Goliaei, M Holzer and AC McHardy introduces a web tool, CoVerage that can identify potential Variants of Interest (pVOIs) using epidemiological frequency data, analysis of changes to the spike protein, and potential for antigenic effects.

The pipeline relies on existing frequency data. In some regions, this may be limited (as mentioned in paragraph that starts at 380), but could be mentioned that this may be especially affected with decreased widespread testing. Additionally, the method requires sufficient time for the pVOI to increase in frequency to meet the designated threshold. This method likely reduces the number of false hits, but may catch pVOIs later than a purely sequence-based method. However, I see the importance of having additional criteria to sort through which variants are likely of concern. I think it would be helpful to be a bit more explicit in the discussion about the benefits and drawbacks of the lineage frequency data. Particularly, you do explain the benefits of looking at pVOIs identified simultaneously in multiple countries (lines 380-389), but what about the potential drawbacks of limiting to those that are? This method might discount a pVOI that is rising in frequency in one location but not yet in others due to a lack of viral exchange. I think this assumption is fine, but it may be worth mentioning when and how things could get missed early with this methodology. Additionally, I think it's important to make it clear in the Abstract and Introduction sections that CoVerage utilizes frequency data.

It is unclear if the model used to identify new pVOIs was informed by the patterns observed in the Omicron variant. If it is, it would be worth mentioning since CoVerage's application is demonstrated using the Omicron variant's appearance. It is useful to use Omicron's pattern to inform the frequency thresholds, amino acid changes of interest, etc, but testing the model with an event that informed the model would not expose biases and potential ways the model could be incomplete in the future variants it can detect. I think it's worth discussing this pitfall. Even better, validating with another strain emergence could help.

Additionally, it would be worth being more explicit about the limitations introduced by focusing on mutations identified on the spike protein. I think it's appropriate to limit the hits by mutation type to reduce false hits, but is also worth discussing. Particularly, what types of mutations is CoVerage ignoring that could affect whether something is a pVOI because its limited to these specific types of mutations (amino acid on spike and antigenic scores) as opposed to all mutations? I assume the use of amino acid changes as opposed to RNA changes is to further focus on changes that physically affect the spike protein. However, there could be mutations that do not physically affect the encoded amino acid but instead affect the regulation of protein production.

Additionally, it is also unclear how the lineages are categorized/classified. This work would benefit from a more detailed explanation of how new lineages are distinguished. I assume the lineages are determined by a threshold of counted amino acid or RNA changes, which would include mutations on the spike protein. What is the threshold? Is it possible this method ends up "double counting" these types of mutations because these mutations are incorporated in the identification of the lineage? One could imagine that this could potentially over-emphasizing the importance of these mutations in the identification of pVOIs.

I appreciate how this work marries epidemiological and phylogenetic approaches. In general, the phylogenetic pipeline looks appropriate. One question I had was, is using the same cost for all amino acid changes and indels in the ancestral sequence reconstruction appropriate here? Additionally, perhaps some protein structural analyses could be added to enhance the argument.

My main concern is with the assessment of antigenic alterations/antigenicity scoring (results on lines 407-441 and methodology on lines 514-522). How are these lineage-specific antigenic alteration scores calculated? What are the thresholds for high vs low antigenic scores? On Line 138 "3.85" was mentioned as a threshold based on being "substantial", but how is this defined? Is this a standardized threshold or is it based on the scores found in this project for the different lineages of Sars-CoV-2? Relying on scores of existing events could potentially be an issue if there is a lot of mutational space that was or was not yet explored by these lineages.

I appreciate the difficult problem of reducing false hits while avoiding missing important potential pVOIs, and I think the methodology described here struck an appropriate balance. I think this work is an appropriate addition to the set of tools used for VOI detection but should be taken in context. Lines 391-395 and 443-459 helped elucidate the context and novelty of the work. Largely, I think it would benefit from more details about the methodology and explanations of the specific choices made.

Additionally, I believe the sentence that starts on line 490 has a grammatical error.

(Remarks on code availability)

Version 2:

Reviewer comments:

Reviewer #1

(Remarks to the Author)

The authors have made substantial revisions to address my major concerns from the previous round, including: (1) better comparison of their work with that of others, (2) enhanced clarity around the methodology for antigenic alteration analysis, (3) revisions to how they report the timing of variant identification with respect to WHO announcements, and (4) a more thorough evaluation of their method's effectiveness, including sensitivity and specificity. I thank the authors for their efforts in addressing these points.

Despite these improvements, I still have a few major concerns about the revised analysis:

Major Comments

1. Validation of antigenic alteration scores

- The improvement in correlation between the authors' antigenic alteration score and experimental data compared to a null model (null $\rho = 0.848$ vs. best model $\rho = 0.888$) appears statistically insignificant given the small dataset (19 variants). With so few data points, this improvement may not be robust.
- The model performance is also counter-intuitive. For instance, the "weights at antigenic sites" model performs worse than the "weights at all sites" model. However, the manuscript does not clarify which positions are referred to as "defined antigenic sites." This ambiguity needs to be addressed.
- A more fundamental concern remains largely unaddressed: while using patterns of amino acid changes in the hemagglutinin (HA) protein of influenza A as a framework for SARS-CoV-2 spike is an innovative idea, it is unclear why this approach should work given the biological differences between these viruses (e.g., structural and functional differences between HA and spike, glycosylation patterns, antigenic assay differences, and host immune responses). The authors should provide more evidence or rationale for why this framework is applicable.

2. Performance evaluation and variant mislabeling

- Several variants appear mislabelled in Figure 3. For example, in panel b, BA.1 (B.1.1.529.1*) and sublineages of Delta (B.1.617.2*) are labeled as "VUM," despite being VOCs according to WHO designations before December 2021. This mislabeling has significant downstream impacts on the evaluation of model performance. For instance, the authors claim that 10 of the 12 lineages not captured by their method were VUMs, which are of "lower public health relevance." Was Omicron BA.1 one of these lineages? This issue undermines the credibility of the performance metrics presented in Figure 3.

3. Length and Verbosity

- The manuscript is lengthy and verbose, especially in the discussion and case studies. While the additional methodological details are appreciated, the authors should aim for conciseness to improve clarity. For example, the discussion could focus on the most critical points, and case studies could be more succinctly presented to highlight key aspects.

Minor Comments

- Line 48: VOC and WHO abbreviations were already defined on lines 36–37.
- Lines 86–87: Clarify what GISAID metadata is referred to (e.g., amino acid substitutions/deletions?). Where in the paper are WHO case numbers used?
- Lines 91–93: Rephrase for clarity. The allele dynamics analysis also requires genome sequence data, not just metadata. How does RKI metadata differ from GISAID in this context?
- Line 109: What is the added value of performing MSA and ancestral sequence reconstruction for mutation frequency analysis compared to simpler methods like those on platforms such as CoV-Spectrum?
- Line 135: Should reference Figure 7 instead of 6.
- Lines 147–148: Asterisks are mentioned in the text but not visible in the heatmaps, only in the lineage dynamics plots. Using asterisks for both antigenic alterations and statistically significant frequency increases is potentially confusing—are their timings distinct?
- Antigenic data source confirmation: Can the authors confirm the data sources (e.g., Table S1 from the Wang et al. preprint for antigenic cartography and Supplementary Data File 2 from the Cox et al. for mean mFRN)?
- Lines 181–182: Specify which amino acid sites are being referred to. This information is missing from the Zenodo supplementary files as well.
- Line 188 and 769: GitHub links do not work.
- Line 200: The sentence appears incomplete.
- Line 210: Alpha is not an "early" lineage compared to Beta. Beta emerged earlier in mid-2020 (see <https://www.nature.com/articles/s41579-023-00878-2>).

- Line 273: Could statistical significance (e.g., Wilcoxon test) be tested for differences between designated and non-designated pVOIs?
- Figure 5: Excellent visualization!

(Remarks on code availability)

For data: The two links provided by the authors appear to be non-functional:

https://github.com/hzi769bifo/corona_lineage_dynamics/tree/main/data and
https://github.com/hzi770bifo/Corona_Variant_Scoring/tree/main/data

For code: The first link (https://github.com/hzi-bifo/Corona_Variant_Scoring) is functional, and the code for the analysis is available with clear instructions. However, the second link (https://github.com/hzi-775bifo/corona_protein_dynamics) does not appear to be working.

Reviewer #2

(Remarks to the Author)

I believe the authors appropriately addressed my comments and concerns. I hope that this work will help identify potential VOIs.

(Remarks on code availability)

The link in the README file (https://github.com/hzi-bifo/Corona_Variant_Scoring/blob/main/README.md) to the BIFO servers is broken in the following quote...

"Required Environment

The analysis pipeline is designed to run on BIFO servers, specifically along with the Sarscoverage environment that can be found here: https://github.com/hzi-bifo/coverage/tree/main/generate_web_data "

I downloaded and attempted to test https://github.com/hzi-bifo/corona_lineage_dynamic using the directions in the README file on the simulated data but unfortunately, I ran into some issues with dependencies on my machine.

Version 3:

Reviewer comments:

Reviewer #1

(Remarks to the Author)

I have no further comments.

(Remarks on code availability)

Regarding the code, I initially only checked that the necessary documents were present, but I have now attempted to run the pipelines for both `corona_lineage_dynamics` and `S-protein dynamics`. I encountered errors in both:

- `corona_lineage_dynamics`: Running `SDPlots_lineages_local.sh` resulted in a connection timeout error when attempting to retrieve alias and lineage date files via R (`get_alias_list.R`):
 Error in functon(type, msg, asError = TRUE) :
 Connection timed out after 300040 milliseconds
 Calls: getURL -> curlPerform -> <Anonymous> -> fun
 Execution halted

- `corona_protein_dynamics`: The provided `environment.yml` file for setting up the conda environment resulted in a "ResolvePackageNotFound" error, preventing installation:

```
conda env create -f environment.yml
Collecting package metadata (repodata.json): done
Solving environment: failed
```

```
ResolvePackageNotFound
```

It would be helpful if the authors could provide clearer instructions for running the code, particularly ensuring all dependencies are properly defined and any external data sources are accessible. That said, I did verify that the actual outputs displayed on their website (sarscoverage.org) appear to be functioning.

Version 4:

Reviewer comments:

Reviewer #1

(Remarks to the Author)

I have no further comments.

(Remarks on code availability)

I was able to run the "corona_lineage_dynamics" pipeline using the pre-downloaded file. It is expected that the pre-downloaded file will also be updated regularly.

Helmholtz Centre for Infection Research | Computational Biology for Infection Research
Inhoffenstraße 7 | 38124 Braunschweig | Germany

Prof. Alice C. McHardy
Head of Department
Computational Biology for Infection Research

Helmholtz Centre for Infection Research)
Inhoffenstrasse 7
38124 Braunschweig

Braunschweig Integrated Centre of Systems Biology
Rebenring 56
38106 Braunschweig

Phone: +49 (0) 531 391-55270
E-mail: alice.mchardy@helmholtz-hzi.de
Web: www.helmholtz-hzi.de/en
[www.tu-braunschweig.de/
forschung/zentren/brics/](http://www.tu-braunschweig.de/forschung/zentren/brics/)

Dear anonymous reviewers,

thank you very much for reviewing our manuscript entitled “***In silico* genomic surveillance by CoVerage predicts and characterizes SARS-CoV-2 Variants of Interest**” and providing constructive comments, which we have found very helpful for further improving the manuscript. We have carefully addressed all the comments by extensively revising the text and performing additional experiments. We made the following changes:

- **Reviewer 1 (Q1)** had concerns about the comparison of CoVerage to other platforms and the depth of description of the available resources. To address this, we compared our antigenic alterations methodology to other techniques that were suggested, such as EVEscape and extended the discussion section to provide a better overview of other resources as well.
- **Both reviewers (R1 Q2 and R2 Q10)** were surprised to see that the methodology for this analysis is still 'in preparation', and the specifics of the scoring system are unclear. We sincerely apologize for not providing this additional manuscript with the first submission. To clarify the details of the method and its effectiveness, we've further extended the methods section of the manuscript and have merged the antigenicity alteration prediction draft manuscript with the CoVerage manuscript for further explanation.
- **Reviewer 1 (Q2 and Q5)** noted the lack of a comprehensive evaluation of the method's sensitivity and specificity regarding the identification of lineages with clear growth advantages. To address this we have added a comprehensive evaluation of the method's performance in terms of identifying VOCs, VOIs and VUMs among predicted pVOIs and antigenically altered lineage predictions, as well as lineages rising to high abundancies.
- **Reviewer 2 (Q2)** was wondering how the frequency-based method implemented in CoVerage relates to a purely sequence-based method in terms of detection performance, asking to be a bit more explicit in the discussion about the benefits and drawbacks, which we addressed, as suggested.

Overall, we are confident that these changes and additional data have substantially improved the manuscript, and we believe that we have addressed the majority of the reviewers' concerns.

Coverage integrates different types of *in silico* analyses of human SARS-CoV-2 evolution and epidemiology to derive relevant information for public health. In a comprehensive assessment of VOIs, VOCs and VUMs identified, we demonstrate how CoVerage can be used to swiftly identify and characterize such variants, with a lead time of almost three months relative to them reaching their maximal abundances. We believe that our work makes a significant contribution to the field, highlighting the potential of data analytics and innovative computational techniques for these purposes. Please do not hesitate to contact us with any additional questions or comments.

Yours sincerely,

Alice McHardy and colleagues

Reviewer 1:

The paper by Norwood et al. introduces CoVerage, a web-based platform designed to monitor the spread of SARS-CoV-2 lineages worldwide and identify mutations in the spike protein that may have antigenic significance. While an automated framework for identifying variants with potential public health implications is beneficial, I am uncertain about the added value of this approach over existing methodologies:

Q1: Comparison with existing methods:

Several existing methods already robustly estimate the growth rate of lineages over time, assess the fitness advantage of mutations, and explore their implications for the antigenic evolution of the virus (e.g., <https://www.science.org/doi/full/10.1126/science.abm1208> and <https://www.science.org/doi/10.1126/scitranslmed.abk3445>). Although CoVerage was compared to some of the online dashboards like NextStrain, CoVariants, and CovidCG, the functionalities of at least some of these platforms were not accurately described in the paper. For example, NextStrain does more than just establish viral lineage nomenclature; it also has features that allow users to assess logistic growth rates and mutational fitness per lineage, features not adequately acknowledged in the comparison.

We thank the reviewer for suggestions and apologize for the omissions. In the revised version, we included these proposed references into the manuscript, in addition to a few others, and discussed them more extensively in both the discussion and introduction sections:

“Though web-based platforms offer various analyses based on publicly available sequencing data that support scientists, public health officials, and the general public in making sense of these highly complex data, there remains a need for methods that identify antigenically altered lineages among the numerous circulating lineages, particularly among lineages rapidly rising in frequency, to support public health related decision making in a timely manner. The early identification of such variants is particularly relevant for vaccine updates to ensure continued vaccine efficacy, such as in the case of Omicron, XBB.1.5 and JN.1¹⁶.”

Additionally, we have extended the discussion to include other features of NextStrain to provide a more comprehensive representation of the state of the art:

“Taken together, CoVerage is a unique web-based resource to identify potential Variants of Interest of SARS-CoV-2 in a timely manner, along with suggesting their degree of antigenic alteration, and alleles of the major surface protein with specific amino acid changes that may provide a selective advantage. Notably, there are other web-based resources to track SARS-CoV-2 variants, among other viruses, and their viral fitness and evolution. NextStrain, for instance, not only established a viral lineage nomenclature based on phylogenetic principles for SARS-CoV-2, but continuously assesses logistic growth rates, immune escape in comparison to BA.2, and mutational fitness per lineage⁶¹. Similarly, PyR0 is a hierarchical Bayesian multinomial logistic regression model that detects lineages increasing in prevalence as well as identifies mutations relative to lineage fitness³² and Episcore, predicts which existing amino acid mutations might contribute to future SARS-CoV-2 VOC's⁶², though neither is run continuously nor available as a web-based platform. Other web-based platforms, such as CoVariants, provide an overview of variant frequencies and shared amino acid mutations⁶³, and CovidCG tracks viral mutations, lineages, and clades in different countries over time⁶⁴. CoVRadar focuses on mutation frequency by location and mutation distribution

among sequences for the molecular surveillance of the SARS-CoV-2 spike protein⁶⁵. Outbreak.info also provides information about lineages and amino acid changes while also reporting prevalence of variants, their geographical distributions and comparisons of changes between lineages⁴⁹.

CoVerage remains unique in its application as it continuously monitors for variants with a potential selective advantage for further study. Only the EVEscape platform developed by Thadani and colleagues, similarly scores emerging variants by their immune escape potential³⁰, however, does not directly identify variants with a potential selective advantage irrespective of this. Additionally, when compared to both mFRN values and antigenic distances, EVEscape had a weaker correlation in comparison to CoVerage antigenic alteration scores. SpikePro on the other hand, correlated similarly to CoVerage scores, but does not continuously score circulating lineages as CoVerage does³¹. Only CoVerage identifies variants with a potential selective advantage using lineage frequency dynamics in combination with predictions of lineage antigenic alterations. This allowed it to accurately identify 31 out of 35 (89%) VOCs and VOIs designated by the WHO with an average lead time of 84 days. We demonstrate the value of this approach in our comprehensive assessment of the value of such predictions for the early identification of past VOCs, VOIs and VUMs, with a lead time of almost three months relative to their maximal abundances and a precision of 79% and recall of 73%. As such, CoVerage lineage predictions are a valuable source of information for further investigations in clinical and epidemiological studies, allowing to prioritize lineages regarding their potential relevance for public health decision making and vaccine updates. The identification of amino acid changes throughout the spike protein that may have selective advantage can further inform antibody design and help understand the molecular basis of adaptive evolution²³. Additionally, CoVerage links to alternative web-based resources for additional information on these selected lineages, providing a comprehensive resource for lineage surveillance. Each of the different analyses provided on CoVerage offers its own benefits when used individually and more so in combination with one another.”

Furthermore, we added an additional section comparing our technique to both EVEscape and SpikePro, alternative fitness scoring methods:

“Neutralization assays allow systematic comparisons of the ability of viral variants, or lineages relative to the wild type virus, regarding their ability to infect cells in the presence of a monoclonal antibody. The fold reduction in neutralization (FRN) is determined by quantifying the concentration of a monoclonal antibody required to prevent infection of cells by a virus or pseudovirus with the spike protein of a particular SARS-CoV-2 variant and comparing these to the results for the wild type sequences, such as the Wuhan-Hu-1 reference sequence using the same experimental conditions. This allows for the comparison of studies even when using different experimental protocols¹¹. Alternatively, the antigenic cartography distances were derived from both human and hamster sera and measured the antigenic distance between the D614G variant and a selected number of VOCs, reflective of the antigenic difference between the two strains²⁵.”

Which was then continued in a later results section: “We also compared this scoring methodology with the scoring results of both EVEscape and SpikePro, alternative viral fitness scoring programs, to antigenic distances available from the Wang et al., 2024 preprint, as well as averaged mean fold reduction in neutralization (mFRN) values for known VOC’s provided by Cox et al., 2022^{11,25,30,31}. EVEscape is a deep learning model that combines fitness predictions with biophysical and structural information to quantify the viral escape potential of SARS-CoV-2 mutations and strains, while SpikePro scores potential fitness of SARS-CoV-2 lineages based on predicted host ACE-2 receptor cell binding affinity, spike protein stability, and mAb binding affinity (**Supplementary Fig. 2**)^{30,31}. When these averaged EVEscape scores were compared alongside averaged antigenic alteration scores calculated by CoVerage from January 2020 to December 2023 to both the antigenic cartography distances as well as averaged mFRN values for known

VOC's using Spearman's correlation coefficient, the antigenic scores produced by CoVerage correlated more strongly with both the antigenic distances and the mFRN values (**Supplementary Fig. 2a**). Specifically, the median CoVerage antigenic alteration scores had a rho 0.019 and 0.1 greater for the antigenic cartography distances and averaged mFRN values respectively, serving as evidence of our antigenic alteration scoring methodology providing even more accurate antigenicity estimates than the recent EVerage technique. Alternatively, the antigenic scores produced by CoVerage correlated more strongly with the antigenic distances than SpikePro with a difference in rho values of 0.007, while the SpikePro scores correlated more strongly with the mFRN values with a difference in rho values of 0.106 (**Supplementary Fig 2b**). Both methods performed similarly, with the difference that SpikePro scored Delta higher than Alpha (with scores of 6.1 and 6.9 respectively) and the two EG lineages, EG.5 and EG.5.1 (both with a score of 13), lower than the XBB lineages, XBB.1.5 and XBB.1.16 (15 and 17 respectively).”

Supplementary Fig. 2: Comparison of predicted antigenic alteration scores and averaged EVEscape scores for VOCs circulating between January 2020 to December 2023 to a antigenic distances from both human and hamster sera and the averaged mFRN values EVEscape score relative to antigenic distance. **b** Comparison of predicted antigenic alteration scores and SpikePro scores for VOCs circulating between January 2020 and December 2023 against the antigenic distances and mFRN values.

Q2: Antigenic alteration analysis:

A major feature of CoVerage that the authors emphasized was its ability to evaluate antigenic lineage alterations over time. However, I was surprised to see that the methodology for this analysis is still 'in preparation', and the specifics of the scoring system are unclear. The examples provided, such as the identification of the S:N501Y mutation in various variants, do not convincingly demonstrate the method's effectiveness, especially given the known saltatory evolution of SARS-CoV-2 and the emergence of highly divergent variants like the Omicron BA.2 and BA.5 lineages will usually lead to noticeable changes in the virus phenotype (see <https://www.nature.com/articles/s41579-023-00878-2> and <https://www.nature.com/articles/s41579-022-00841-7>).

We sincerely apologize for not providing this additional manuscript in preparation with the first submission. To further clarify the method and its effectiveness, we've further extended the methods section of the manuscript and have merged the antigenicity alteration prediction manuscript with the CoVerage manuscript for a more expansive explanation of the method. The revised methods section is outlined below:

Fig. 7: Schematic outline of the technique to predict the antigenic alteration of SARS-CoV-2 spike protein sequences using a IAV-derived viral immune escape matrix. **a** The inputs for antigenic tree inference are Hemagglutination Inhibition Assay (HIA) data table for IAV and a genealogy inferred from isolate sequences of the major viral surface protein, and the corresponding amino acid sequences²³ **b** Antigenic tree with antigenic branch weights inferred and amino acid changes in the viral antigenic reconstructed for the evolutionary branches. Values such as '0/2' indicate antigenic up and down branch weights, reflecting the asymmetric nature of hemagglutination inhibition data, where antigenic distance from a viral antigen strain *a* to antiserum raised against another strain *b* can be different from the distance of an antiserum raised against antigen *a* to an antigen *b*, respectively. The red color traced through the IAV Antigenic Tree indicates an example path between the A1 and A3 variant, for which the sum of edge weights correspond to the red values for A1 - A3 and purple values for A3 - A1 in the HIA table in panel a Several changes in bold are exemplary shown in the IAV Antigenic Tree table, which represents for every position-specific amino acid mapped to a tree branch the antigenic branch weights. The up/down weights inferred

in this tree are then compiled into **c** position-agnostic viral immune evasion antigenic alteration matrices for pairs of amino acids, shown here with the example of averaging over inferred up and down weights. **d** These weights are then used to score amino acid changes across the spike protein of SARS-CoV-2 to calculate sequence-specific antigenic alteration scores, which are then combined in lineage and country-wise alteration scores.

“We quantify the degree of antigenic divergence of a viral isolate relative to another one using an antigenic scoring matrix. The antigenic scoring matrix includes antigenic alteration estimates for pairs of amino acid changes derived from a previously inferred antigenic tree for Influenza A viruses (IAV)²³. These were established by inferring antigenic weights, i.e. antigenic branch lengths, from hemagglutination inhibition (HI) assay data onto a maximum likelihood phylogenetic tree and matching these to branch-specific sets of amino acid changes inferred by ancestral character state reconstruction for the major viral surface protein for human influenza A (H3N2) viruses, combining genetic with evolutionary and serological data (Fig. 7a,b). The inference of the antigenic tree is based on a least-squares optimization (LSO) procedure, which fits dissimilarities derived from HI assay data onto an Influenza A virus (IAV) evolutionary tree (Fig. 7a)²³. The branch length is then representative of the antigenic difference between nodes in the genealogy, which, after reconstructing amino acid changes via ancestral character state reconstruction and mapping these onto the tree branches throughout the tree, indicates the potential impact of the branch-associated amino acid changes on the immune escape of the virus. As HI assay data is asymmetric, the antigenic distance from a viral antigen strain a to antiserum raised against another strain b can be different from the distance of an antiserum raised against antigen a to an antigen b, respectively. The antigenic tree uses both up and down weights for each branch. Here, the up weights are representative of the antigenic distance from isolates below the branch to all other isolates and the down weights are representative of the distances from isolates outside the subtree to isolates below the branch²³. It is this combination of up/down weights and amino acid changes throughout the inferred IAV antigenic tree (Fig. 7b) that is used to create a viral immune escape matrix (Fig. 7c), which includes one antigenic alteration weight for each amino acid change. These weights can then be applied to amino acid changes occurring across the SARS-CoV-2 spike protein and quantify the antigenic alteration of SARS-CoV-2 strains relative to the original Wuhan-Hu-1 strain (Fig. 7d).

From the inferred IAV antigenic tree, several antigenic alteration matrices specifying weights for all pairs with amino acid changes with evidence in the antigenic tree were defined. The weights serve as a measure for the potential impact that the amino acid substitution has on the antigenicity of the variant. $M_{ant_alt_0}$ includes weights averaging over the respective up or down weights seen for a particular change across all positions, such as the change N_K, assigned a weight of 0.5, averaging over $(0+0+0+2)/4$ (Fig 7c). Matrix $M_{ant_alt_1}$ includes weights from $M_{ant_alt_0}$ averaged across change and reverse change. The matrix $M_{ant_alt_2}$ includes a total of forty-one weights, and matrix $M_{ant_alt_2}$ includes antigenic weights only for changes occurring at least three times in the antigenic tree. All unspecified pairs of changes in the antigenic matrices include missing values and are not counted.

We then explored whether antigenic weights for pairs of amino acid changes derived from data for the antigenic evolution of human influenza A (H3N2) viruses are predictive for the degree of antigenic evolution of human SARS-CoV-2 viruses. Specifically, the antigenic alteration score for a lineage l is calculated to predict the antigenic alterations of each SARS-CoV-2 spike protein sequence thereof relative to the Wuhan-Hu-1 strain. To this end, antigenic weights from the viral immune escape matrix are applied to the amino acid changes of a lineage throughout the spike protein relative to the Wuhan-Hu-1. The weights are subsequently summed per isolate and averaged across all isolates of a given circulating Pango lineage to calculate the final antigenic score,

For scoring currently circulating SARS-CoV-2 lineages, we used the metadata provided by GISAID and filtered isolates by the most recent complete month (ie. the previous month, or in the Omicron case study by the desired months) based on the isolate collection date. Sequences under review were removed. Among other information, the metadata contains amino acid substitutions or deletions occurring in each isolate as well as their Pango lineage, which would be used to assign an antigenic weight to each change occurring at the previously defined sites. Once assigned to each change, the individual amino acid weights were then summed per isolate and then averaged across each Pango lineage occurring in the selected month to assign one final antigenic score to each circulating Pango lineage that month, such that:

$$a_l = \frac{1}{n_{il}} \sum w_{aa}$$

where a_l is the antigenic score for the Pango lineage l and n_{il} represents the number of isolates per Pango lineage in the given month and w_{aa} represents the assigned weight for an amino acid change.

In addition to the individual lineage scores, a country-wide antigenic alteration score, a_c , is calculated by summing over all antigenic lineage scores, weighted by their frequencies for a particular period in time in a country c :

$$a_c = \sum \left(a_l \times \frac{n_{ilc}}{n_{ic}} \right)$$

Here, n_{ilc} is the number of isolates of each Pango lineage for country c and n_{ic} is the total number of isolates per country.”

We include below the results from our comprehensive evaluation of the methods’ merits using published serological data from multiple studies and quantitative evaluation of the methods’ ability for the timely identification of VOIs, VOCs and VUMs.

“To assess antigenic alteration predictions for SARS-CoV-2 lineages, we performed a thorough benchmarking versus experimentally measured antigenic cartography distances²⁵ and averaged mean fold reduction in neutralization (mFRN)¹¹. Here we considered a comprehensive set of previously circulating and more recent lineages including Alpha, Beta, Delta, Gamma, Epsilon, Lambda, Mu, Omicron, and subsequent Omicron sublineages.

Neutralization assays allow systematic comparisons of the ability of viral variants, or lineages relative to the wild type virus, regarding their ability to infect cells in the presence of a monoclonal antibody. The fold reduction in neutralization (FRN) is determined by quantifying the concentration of a monoclonal antibody required to prevent infection of cells by a virus or pseudovirus with the spike protein of a particular SARS-CoV-2 variant and comparing these to the results for the wild type sequences, such as the Wuhan-Hu-1 reference sequence using the same experimental conditions. This allows for the comparison of studies even when using different experimental protocols¹¹. Alternatively, the antigenic cartography distances were derived from both human and hamster sera and measured the antigenic distance between the D614G variant and a selected number of VOCs, reflective of the antigenic difference between the two strains²⁵.

Fig. 2: Comparison of predicted lineage antigenic alteration scores (AAS) versus experimentally determined distances from antigenic cartography (panels a, c, e, g, i, k) and mean mFRN values for all mAbs (panels b, d, f, h, j, l). The y-axes represent the predicted antigenic alteration scores of variants relative to the Wuhan Strain using data across the spike protein from January 2020 to December 2023. The x-axes represent either the antigenic cartography distance²⁵ or the mean mFRN values¹¹, respectively, with their correlation assessed using Spearman's correlation coefficient 'rho'. **a, b** Antigenic alteration scores were calculated as the number of changes at defined antigenic sites; **c, d** with antigenic weights at all altered sites on the spike protein; **e, f** With antigenic weights at defined antigenic sites on the spike protein; **g, h** without weights at all sites on the spike protein; **i, j** Using weights at all sites but without directionality of the amino acid change and **k, l** Averaging over antigenic weights from the Influenza A antigenic tree that occurred at least three times at all altered sites.

We analyzed different ways of scoring the antigenic alterations of lineages based on their changes relative to the original Wuhan strain and assessed their value using the experimental data. The evaluated scoring methods include (1) scoring antigenic alterations at amino acid sites of the spike protein that we identified from literature during the first year of the pandemic using *M_ant_alt_0*, (2) scoring antigenic alterations across all amino acid sites of the spike protein, (3) simply counting amino acid changes that occur at the sites defined in method one, (4) counting amino acid changes across the spike protein, (5) using an alternative scoring matrix, *M_ant_alt_1*, with antigenic weights averaged across change and reverse change, respectively on all altered sites for the spike protein, and finally, (6) using an alternative scoring matrix, *M_ant_alt_2*, which included averaged antigenic weights from the Influenza A antigenic tree for amino acid changes that occur at least three times across the tree (Methods; https://github.com/hzi-bifo/Corona_Variant_Scoring).

Based on these results, the scoring method that utilized the weights from the universal immune matrix at all amino acid sites across the spike protein correlated the best with the antigenic cartography distances with a rho of 0.888 (**Fig. 2c**) and the method utilizing the bidirectional weights correlated the best with the mFRN values with a rho of 0.917. The bidirectional weights method also performed similarly to the method using weights at all amino acid sites when compared to the antigenic cartography distances with a 0.001 difference in the rho values. Utilizing the weights at all amino acid changes across the spike protein as well as using the bidirectional weights were the only two methods with better correlation than the method counting the amino acid changes across the spike protein (without weights at all amino acid changes; **Fig. 2g,h**), which we used as a baseline measure. Hence the weights serve as an important addition to the scoring method, though the application at known antigenic sites may not.

In all tested methods, except for the method without weights at antigenic sites and with the weights defined by three occurrence throughout the IAV antigenic tree, both the Beta (Pango lineage B.1.351) and the Gamma (Pango lineage P.1) variant had a higher antigenic alterations score relative to both Alpha and Delta, Alpha being the predominant lineage circulating in early 2021 and Delta being the predominant lineage circulating prior to the rise of Omicron in late 2021 (**Fig. 2**)²⁶. Both the Beta and Gamma lineage are antigenically similar to one another, with Beta having higher cross neutralization titers against Gamma than other variants, potentially due to their shared mutation at K417^{27,28}. Additionally, with each of the methods tested there was a distinguishable increase in the antigenic alteration scores between earlier lineages (Alpha, Epsilon, Lambda), to intermediate pre-Omicron lineages (Gamma and Beta), and then to Omicron and its subsequent lineages. This was also reflected in both the antigenic cartography distances as well as averaged mFRN values as Omicron has been proven to be antigenically distinct from prior lineages with an increased immune escape capacity from ancestral variant vaccinated sera²⁷.

Overall, each of the methods correlated strongly with both the antigenic distances and averaged mFRN values, which were used as a proxy measure of immune escape capacity in this comparison. Only two methods (**Fig. 2c,d** and **Fig. 2i,j**) have a greater correlation than the established baseline method, considering counts of changes across the spike protein (**Fig. 2g,h**). With this, we selected the method that applied immune escape weights at all amino acid changes across the spike protein (**Fig. 2c,d**) to progress with, as the differences between the two correlations of each method were insignificant after determining the difference between the z transformed correlations (correlation *r.test* R programming psych package; z-value 0.02 for antigenic cartography correlation with 0.98 probability and z-value 0.17 for mFRN values with 0.87 probability)²⁹. By utilizing the method with weights from the viral immune escape matrix at all amino acid sites there was no inference of potential antigenic weights for reverse amino acid changes to those found in the original IAV antigenic tree, leaving room for directionality of changes to be considered in the model.”

Q3: Timing of variant identification by CoVerage vs. WHO announcements:

The emphasis by the authors on the timing of the identification of variants by CoVerage compared to WHO announcements can be misleading. WHO decisions are based on sequence data and epidemiological evidence available at the time, often much delayed from collection to submission to GISAID or other online platforms. Stating that CoVerage identified Omicron before its official designation by WHO overlooks the swift response by Southern African scientists that led to its rapid identification as a VOC. In fact, Omicron was indeed a good example of a swift announcement from WHO about its designation as a VOC thanks to the work of the Southern African scientists.

We fully agree with the reviewer in that Omicron has been a great example of a rapid response of the WHO. We have adapted the respective statement in the discussion section to:

“Consistent with the rapid designation of Omicron as a VOC by the WHO, which in addition to sequencing information also considers epidemiological evidence, CoVerage identified this pVOI from the submitted data just three days before.”

And finally, we clarified that pVOIs are intended to serve as additional evidence for decisions made by public health decision bodies, rather than a competing effort in the discussion section:

“We demonstrate the value of this approach in our comprehensive assessment of the value of such predictions for the early identification of past VOCs, VOIs and VUMs, with a lead time of almost three months relative to their maximal abundances and a precision of 79% and recall of 73%. As such, CoVerage lineage predictions are a valuable source of information for further investigations in clinical and epidemiological studies, allowing to prioritize lineages regarding their potential relevance for public health decision making and vaccine updates. The identification of amino acid changes throughout the spike protein that may have selective advantage can further inform antibody design and help understand the molecular basis of adaptive evolution²³. Additionally, CoVerage links to alternative web-based resources for additional information on these selected lineages, providing a comprehensive resource for lineage surveillance. Each of the different analyses provided on CoVerage offers its own benefits when used individually and more so in combination with one another.”

Q4: On a separate note, the term 'potential variant of interest (pVOI)' used in the paper could contribute to confusion, as it lacks a clear definition and differs from established WHO terminology. A term like 'variant under monitoring' might align better with existing nomenclature without adding ambiguity (see <https://www.who.int/publications/m/item/updated-working-definitions-and-primary-actions-for--sars-cov-2-variants>).

We appreciate the thoughtful and constructive suggestion. To clarify these distinctions, we further extended the parts in the Introduction, and checked the official definition of the pVOIs, specifying further details on all parameters used by the pVOI prediction method, as outlined below.

Revised sentence in the introduction:

“The term pVOI, adapted from the WHO defined Variant of Interest (VOI), defines potential Variants of Interest (pVOI) as predicted by CoVerage, which identifies pVOIs as those that increase significantly in frequency over time and rise above a predominance threshold for the first time in a fully automated fashion¹⁷”

The exact definition of a pVOI in the results section has been expanded with further parameters (page 5, line 111 to 132):

“For the analysis of lineage dynamics and to identify pVOIs, i.e. lineages that may have a selective advantage to spread in the host population, as well as for identifying sets of amino acid changes on the spike protein rapidly increasing in frequency, we adapted a technique that we developed for recommending vaccine strain updates for the seasonal influenza vaccine (<https://github.com/hzi-bifo/SDplots>)²⁰. There are three methodologies for identifying pVOI lineages: the ‘standard’ method identifies lineages increasing significantly in frequency over time with monthly intervals using Fisher’s exact test. The sub-lineage corrected method for pVOI identification includes genome sequence assignments belonging to sublineages in the pVOI assessment. Finally, the sliding window method uses a windowed time period for the analysis to achieve a more fine-grained analysis of shorter time intervals (Methods).

For the lineage dynamics analyses, Pango lineages were used as nomenclature, which define epidemiologically relevant phylogenetic clusters, in which new lineages are only designated if the lineage has high coverage and contains a sufficient number of sequences²¹. All methods utilize Fisher’s exact test to identify lineages that are significantly increasing in frequency over two consecutive time intervals and correct for multiple testing using the false discovery rate (FDR, $\alpha = 0.05$)²². This identifies pVOIs as lineages that are both significantly on the rise and increase above a predominance threshold of 0.1 within the same time frame. Notably, lineages may also be falsely identified as pVOIs due to unrepresentative sampling, e.g., data biases towards certain areas, large clonal outbreaks, or population bottlenecks. Therefore, identified pVOIs should be evaluated carefully, e.g. in combination with epidemiological data and experimental evidence showing that amino acid changes and altered positions observed in a pVOI lineage are likely to confer a selective advantage.”

We also added additional information to the discussion:

“The pVOIs and antigenically altered variants identified by CoVerage thus comprise a somewhat larger, but highly informative set of variants with a potential selective advantage for identifying the VOIs, VOCs and VUMs defined by the WHO. The latter are required to show early or confirmed signs of a growth rate advantage, together with further criteria such as the presence of genetic changes that are suspected,

predicted or known to affect virus characteristics, along with increasing case numbers or other apparent epidemiological impacts indicating an increasing public health risk, as confirmed by a panel of experts, while the latter contains genetic changes suspected to affect viral characteristics with early signals of growth advantage but either phenotypic or epidemiological impact remains unclear⁵⁸. CoVerage predictions are fully reproducible as they are derived from a defined set of input data, with a fully deterministic, statistical assessment of globally available data, to support and facilitate further expert assessments.”

Q5: Evaluation of method effectiveness:

It seems to me that the paper selectively presents instances where CoVerage successfully identified lineages and mutations with clear growth advantages. And there is a clear lack of a comprehensive evaluation of the method’s sensitivity and specificity. The frequency of false positives, where variants are incorrectly labeled as pVOIs, is not discussed, which is crucial for assessing the reliability of the platform in early and reliable detection of variants that need to be monitored.

We apologize for not providing further comprehensive information on this in the prior version. We have addressed this as suggested, adding a comprehensive evaluation of the method's performance in identifying VOCs, VOIs and VUMs among predicted pVOIs and antigenically altered lineage predictions.

“Widely spreading antigenically altered SARS-CoV-2 lineages can be detected early

For validation of the pVOI identification and antigenically altered scoring methodologies created by CoVerage, we determined, for pVOIs and antigenically altered lineages predicted until the end of June 2024, the time it took for these lineages to reach their peak frequencies (Fig. 3). The frequency for each lineage in each country was calculated using a one week sliding window with a step size of two days. From here, the peak sequence count, the corresponding frequency, and the date of the peak were identified. To ensure robustness and mitigate bias from rare lineages or inadequate sequence sampling, we included only those records where the peak sequence count was at least 5 within the respective window. In our comprehensive evaluation, a total of 404 detected pVOIs from 91 countries were analyzed, corresponding to 1360 pVOI predictions for individual countries. On average, CoVerage identified the pVOI using data from 49 days earlier than the date when it reached its peak frequency. For the countries with more data available (peak sequence count ≥ 100 for the respective lineage), this lead time extended to 78 days (Fig. 3). The antigenically altered pVOIs have a day difference of 72 compared to 24 for non-altered pVOIs and reached a much higher peak frequency than the non-altered ones (Fig. 3a), indicating the relevance of antigenic alterations for providing a selective advantage to variant lineages.

Overall, 320 pVOI lineages identified by CoVerage belong to the 45 officially designated VOCs, VOIs, and VUMs, covering 33 of these designated lineages, resulting in a precision of 79% (320/404) and a recall of 73% (33/45). Among the 12 lineages not captured by pVOIs, 10 are VUMs, thus the lowest category of relevance from public health monitoring. No VOCs were missed. On average, these pVOIs were first identified using data collected 84 days before their official WHO designation. The pVOIs not designated by WHO were primarily identified in 2020, during the early stages of the pandemic, when available sequence data was oftentimes limited and less from systematic representative surveillance efforts, as in the following years (Fig. 3b). Notably, the antigenic scores and standardized scores of pVOIs that were also WHO designated lineages were much higher than those of the non-designated pVOIs, and increasing in order of VUMs, VOIs and VOCs, in line with their proven relevance for public health decision making (Fig. 3c). To

evaluate the benefits of our method relative to a sequence analysis/based method, we compared PyR0³² relative to the antigenic scoring method (**Fig. 3d**), which showed that the antigenic alteration score of variants of CoVerage predicted VOCs, VOIs, and VUMs more accurately than the R/R_A score used by PyR0, with an Area Under the Precision-Recall Curve (PRAUC) of 0.84 vs 0.74, respectively. Overall, these findings underscore our method's capability to effectively predict the ascent of lineages relevant for public health decision making with a growth advantage well ahead of their peak frequency, providing critical lead time for public health interventions.”

“Fig. 3: Performance evaluation of CoVerage. a The difference in days and frequency of pVOI identification of lineages relative to their peak frequencies. The dots in red indicate predicted antigenically altered pVOIs, while the turquoise dots represent predicted non-altered pVOIs. In the side boxplot, the mean is indicated by the black diamond and the number above it. **b** Antigenic scores of pVOIs identified

using data before December 2021. Colors represent different WHO designations. The x-axis shows the month of identification, with "Apr. 20" indicating April 2020. The y-axis displays the $-\log_{10}$ transformed p-value of pVOI detection. The size of the points represents the antigenic score of the pVOI. **c** Standardized (z-score) antigenic scores for variants and their sublineages for different WHO variant categories designations until June 2024. Here the horizontal bar within the box indicates the median, while the black diamond and the number above it represents the mean. "NA" indicates pVOIs not designated by WHO. **d** PRAUC of CoVerage antigenic score and PyR0 R/R_A in classifying/predicting lineages as WHO-designated or non-WHO-designated VUMs, VOIs and VOCs. Data for both methods were collected before December 2021, with the ground truth including WHO-designated lineages up to May 2022 to demonstrate early detection capabilities. In **b** and **c**, "NA" indicates pVOIs not designated by WHO."

Reviewer 2:

Q1: "In silico genomic surveillance by CoVerage predicts and characterizes SARS-CoV-2 Variants of Interest" by K Norwood, ZL Deng, S Reimering, G Robertson, MH Foroughmand-Araabi, S Goliaei, M Holzer and AC McHardy introduces a web tool, CoVerage that can identify potential Variants of Interest (pVOIs) using epidemiological frequency data, analysis of changes to the spike protein, and potential for antigenic effects. The pipeline relies on existing frequency data. In some regions, this may be limited (as mentioned in paragraph that starts at 380), but could be mentioned that this may be especially affected with decreased widespread testing.

Thank you for this helpful suggestion, we have added a statement to the discussion section:

"Detection may be affected if genomic surveillance would be decreased further in the future with the virus becoming endemic."

Q2: Additionally, the method requires sufficient time for the pVOI to increase in frequency to meet the designated threshold. This method likely reduces the number of false hits, but may catch pVOIs later than a purely sequence-based method. However, I see the importance of having additional criteria to sort through which variants are likely of concern. I think it would be helpful to be a bit more explicit in the discussion about the benefits and drawbacks of the lineage frequency data. Particularly, you do explain the benefits of looking at pVOIs identified simultaneously in multiple countries (lines 380-389), but what about the potential drawbacks of limiting to those that are? This method might discount a pVOI that is rising in frequency in one location but not yet in others due to a lack of viral exchange. I think this assumption is fine, but it may be worth mentioning when and how things could get missed early with this methodology.

Thank you for this thoughtful suggestion. To address this, we clarified the respective section, referring to our systematic analysis of VOIs, VOCs and VUM predictions, which evaluated predictions on a per-country basis. We have extended the discussion to include these points, as outlined below:

"All pVOIs identified in one or more countries within the last month reported by CoVerage in the global pVOI heatmaps. A systematic assessment of these pVOI predictions in combinations with their antigenic alteration predictions showed that this accurately identified 89% of the VOIs and VOCs designated by the WHO since the establishment of SARS-CoV-2 in the human population on average almost three months before their peak abundances were reached."

We also added the following to the result section (p. 11 line 274):

“To evaluate the benefits of our method relative to a sequence analysis/based method, we compared PyR0³² relative to the antigenic scoring method (Fig. 3d), which showed that the antigenic alteration score of variants of CoVerage predicted VOCs, VOIs, and VUMs more accurately than the R/R_A score used by PyR0, with an Area Under the Precision-Recall Curve (PRAUC) of 0.84 vs 0.74, respectively.”

Fig. 3: Performance evaluation of CoVerage. **a** The difference in days and frequency of pVOI identification of lineages relative to their peak frequencies. The dots in red indicate predicted antigenically altered pVOIs, while the turquoise dots represent predicted non-altered pVOIs. In the side boxplot, the mean is indicated by the black diamond and the number above it. **b** Antigenic scores of pVOIs identified

using data before December 2021. Colors represent different WHO designations. The x-axis shows the month of identification, with "Apr. 20" indicating April 2020. The y-axis displays the $-\log_{10}$ transformed p-value of pVOI detection. The size of the points represents the antigenic score of the pVOI. **c** Standardized (z-score) antigenic scores for variants and their sublineages for different WHO variant categories designations until June 2024. Here the horizontal bar within the box indicates the median, while the black diamond and the number above it represents the mean. "NA" indicates pVOIs not designated by WHO. **d** PRAUC of CoVerage antigenic score and PyR0 R/R_A in classifying/predicting lineages as WHO-designated or non-WHO-designated VUMs, VOIs and VOCs. Data for both methods were collected before December 2021, with the ground truth including WHO-designated lineages up to May 2022 to demonstrate early detection capabilities. In **b** and **c**, "NA" indicates pVOIs not designated by WHO.

Q3: Additionally, I think it's important to make it clear in the Abstract and Introduction sections that CoVerage utilizes frequency data.

Thank you for this helpful comment. To address this, we have added a sentence in the abstract:

"Here, we describe the CoVerage system (www.sarscoverage.org) for viral genomic surveillance, which continuously predicts and characterizes novel and emerging potential Variants of Interest (pVOIs) from country-wise lineage frequency dynamics together with their antigenic and evolutionary alterations utilizing the GISAID viral genome resource."

In addition we have revised the following in the Introduction:

"The platform implements methods that continuously search for pVOIs from global, country-wise viral variant frequency dynamics and predicts relevant evolutionary changes and their antigenic alterations relative to the original Wuhan strain, using publicly available viral surveillance and genome data in a fully automated, timely fashion."

Q4: It is unclear if the model used to identify new pVOIs was informed by the patterns observed in the Omicron variant. If it is, it would be worth mentioning since CoVerage's application is demonstrated using the Omicron variant's appearance. It is useful to use Omicron's pattern to inform the frequency thresholds, amino acid changes of interest, etc, but testing the model with an event that informed the model would not expose biases and potential ways the model could be incomplete in the future variants it can detect. I think it's worth discussing this pitfall. Even better, validating with another strain emergence could help.

We thank the reviewer for this insightful comment. Omicron served as an independent test case, along with later variants "BQ", "XBB", and "JN.1". To clarify this, we have added a sentence to the respective result section:

"We retrospectively analyzed data from the time of emergence of the Omicron lineage or Pango lineage BA.1 as the Omicron lineage, to assess the detection of pVOIs, prediction of antigenic alterations (Case study 1b), and of relevant amino acid changes (Case study 1c). This was after finalizing all methodological settings of pVOI detection in first year of the pandemic, and utilizing an antigenic alteration scoring method, which scores all amino acid changes in the spike protein relative to the original Wuhan strain, thus not including knowledge about relevant sites for Omicron or other variants."

Q5: Additionally, it would be worth being more explicit about the limitations introduced by focusing on mutations identified on the spike protein. I think it's appropriate to limit the hits by mutation type to reduce false hits, but is also worth discussing. Particularly, what types of mutations is CoVerage ignoring that could affect whether something is a pVOIs because its limited to these specific types of mutations (amino acid on spike and antigenic scores) as opposed to all mutations?

Thank you for this interesting comment. In response, we updated the methodology for the antigenic alteration scoring analysis, to include also a scoring of all all amino acid changes occurring throughout the spike protein, rather than the predefined sites , which indeed turned out to provide a further improvement in predictive performance in comparison to the selected set of sites, demonstrating that immune evasion of the virus was not appropriately covered by the set of sites impacted over the course of the first year of the pandemic. We've outlined these details as an additional results section for a more in depth evaluation:

“To assess antigenic alteration predictions for SARS-CoV-2 lineages, we performed a thorough benchmarking versus experimentally measured antigenic cartography distances²⁵ and averaged mean fold reduction in neutralization (mFRN)¹¹. Here we considered a comprehensive set of previously circulating and more recent lineages including Alpha, Beta, Delta, Gamma, Epsilon, Lambda, Mu, Omicron, and subsequent Omicron sublineages.

Neutralization assays allow systematic comparisons of the ability of viral variants, or lineages relative to the wild type virus, regarding their ability to infect cells in the presence of a monoclonal antibody. The fold reduction in neutralization (FRN) is determined by quantifying the concentration of a monoclonal antibody required to prevent infection of cells by a virus or pseudovirus with the spike protein of a particular SARS-CoV-2 variant and comparing these to the results for the wild type sequences, such as the Wuhan-Hu-1 reference sequence using the same experimental conditions. This allows for the comparison of studies even when using different experimental protocols¹¹. Alternatively, the antigenic cartography distances were derived from both human and hamster sera and measured the antigenic distance between the D614G variant and a selected number of VOCs, reflective of the antigenic difference between the two strains²⁵.

Fig. 2: Comparison of predicted lineage antigenic alteration scores (AAS) versus experimentally determined distances from antigenic cartography (panels a, c, e, g, i, k) and mean mFRN values for all mAbs (panels b, d, f, h, j, l). The y-axes represent the predicted antigenic alteration scores of variants relative to the Wuhan Strain using data across the spike protein from January 2020 to December 2023. The x-axes represent either the antigenic cartography distance²⁵ or the mean mFRN values¹¹, respectively, with their correlation assessed using Spearman's correlation coefficient 'rho'. **a, b** Antigenic alteration scores were calculated as the number of changes at defined antigenic sites; **c, d** with antigenic weights at all altered sites on the spike protein; **e, f** With antigenic weights at defined antigenic sites on the spike protein; **g, h** without weights at all sites on the spike protein; **i, j** Using weights at all sites but without directionality of the amino acid change and **k, l** Averaging over antigenic weights from the Influenza A antigenic tree that occurred at least three times at all altered sites.

We analyzed different ways of scoring the antigenic alterations of lineages based on their changes relative to the original Wuhan strain and assessed their value using the experimental data. The evaluated scoring methods include (1) scoring antigenic alterations at amino acid sites of the spike protein that we identified from literature during the first year of the pandemic using *M_ant_alt_0*, (2) scoring antigenic alterations across all amino acid sites of the spike protein, (3) simply counting amino acid changes that occur at the sites defined in method one, (4) counting amino acid changes across the spike protein, (5) using an alternative scoring matrix, *M_ant_alt_1*, with antigenic weights averaged across change and reverse change, respectively on all altered sites for the spike protein, and finally, (6) using an alternative scoring matrix, *M_ant_alt_2*, which included averaged antigenic weights from the Influenza A antigenic tree for amino acid changes that occur at least three times across the tree (Methods; https://github.com/hzi-bifo/Corona_Variant_Scoring).

Based on these results, the scoring method that utilized the weights from the universal immune matrix at all amino acid sites across the spike protein correlated the best with the antigenic cartography distances with a rho of 0.888 (**Fig. 2c**) and the method utilizing the bidirectional weights correlated the best with the mFRN values with a rho of 0.917. The bidirectional weights method also performed similarly to the method using weights at all amino acid sites when compared to the antigenic cartography distances with a 0.001 difference in the rho values. Utilizing the weights at all amino acid changes across the spike protein as well as using the bidirectional weights were the only two methods with better correlation than the method counting the amino acid changes across the spike protein (without weights at all amino acid changes; **Fig. 2g,h**), which we used as a baseline measure. Hence the weights serve as an important addition to the scoring method, though the application at known antigenic sites may not.

In all tested methods, except for the method without weights at antigenic sites and with the weights defined by three occurrence throughout the IAV antigenic tree, both the Beta (Pango lineage B.1.351) and the Gamma (Pango lineage P.1) variant had a higher antigenic alterations score relative to both Alpha and Delta, Alpha being the predominant lineage circulating in early 2021 and Delta being the predominant lineage circulating prior to the rise of Omicron in late 2021 (**Fig. 2**)²⁶. Both the Beta and Gamma lineage are antigenically similar to one another, with Beta having higher cross neutralization titers against Gamma than other variants, potentially due to their shared mutation at K417^{27,28}. Additionally, with each of the methods tested there was a distinguishable increase in the antigenic alteration scores between earlier lineages (Alpha, Epsilon, Lambda), to intermediate pre-Omicron lineages (Gamma and Beta), and then to Omicron and its subsequent lineages. This was also reflected in both the antigenic cartography distances as well as averaged mFRN values as Omicron has been proven to be antigenically distinct from prior lineages with an increased immune escape capacity from ancestral variant vaccinated sera²⁷.

Overall, each of the methods correlated strongly with both the antigenic distances and averaged mFRN values, which were used as a proxy measure of immune escape capacity in this comparison. Only two methods (**Fig. 2c,d** and **Fig. 2i,j**) have a greater correlation than the established baseline method, considering counts of changes across the spike protein (**Fig. 2.g,h**). With this, we selected the method that applied immune escape weights at all amino acid changes across the spike protein (**Fig. 2c,d**) to progress with, as the differences between the two correlations of each method were insignificant after determining the difference between the z transformed correlations (correlation *r.test* R programming psych package; z-value 0.02 for antigenic cartography correlation with 0.98 probability and z-value 0.17 for mFRN values with 0.87 probability)²⁹. By utilizing the method with weights from the viral immune escape matrix at all amino acid sites there was no inference of potential antigenic weights for reverse amino acid changes to those found in the original IAV antigenic tree, leaving room for directionality of changes to be considered in the model.”

Furthermore, we have updated the discussion regarding why we chose to use the spike protein specifically versus for example a genome-wide scoring of changes:

“CoVerage predicts antigenically altered lineages with a novel method that scores evolutionary changes using an antigenic alteration matrix defined from genotype-to-antigenic phenotype association mappings from long-term evolution of seasonal influenza A (H3N2) viruses. The method assesses all changes in the spike protein, which is the key immunogen and protein for the binding of SARS-CoV-2 to host receptor cells, and as such is a target for vaccines and antibody therapy²⁴. We show that this method in combination with pVOI predictions allows to confidently identify priority variants and their sublineages over the past couple of years, that were defined by the WHO of public health relevance, such as VOCs, VOIs and VUMs, with on average almost three months prior of them reaching their maximum global frequencies.”

Q6: I assume the use of amino acid changes as opposed to RNA changes is to further focus on changes that physically affect the spike protein. However, there could be mutations that do not physically affect the encoded amino acid but instead affect the regulation of protein production.

Thank you for this comment. This is an interesting angle, which would require a substantial revision of the analyses' framework and a systematic, large-scale exploration, which could be a follow-up to this study in the future.

Q7: Additionally, it is also unclear how the lineages are categorized/classified. This work would benefit from a more detailed explanation of how new lineages are distinguished. I assume the lineages are determined by a threshold of counted amino acid or RNA changes, which would include mutations on the spike protein. What is the threshold? Is it possible this method ends up "double counting" these types of mutations because these mutations are incorporated in the identification of the lineage? One could imagine that this could potentially over-emphasizing the importance of these mutations in the identification of pVOIs.

We apologize for the lack of clarity. To address this, we have added the following: In the analysis, we use lineage nomenclature assigned by Pangolin. Rather than defining new lineages, we identify pVOIs based on the lineages already established by Pangolin. When identifying the amino acid changes, spike protein sequences were extracted from genome sequences then were aligned together with the S protein of Wuhan/IPBCAMS-WH-01/2019 as reference. To clarify this, we have extended the method section:

“For the lineage dynamics analyses, Pango lineages were used as nomenclature, which define epidemiologically relevant phylogenetic clusters, in which new lineages are only designated if the lineage has high coverage and contains a sufficient number of sequences²¹.”

Q8: I appreciate how this work marries epidemiological and phylogenetic approaches. In general, the phylogenetic pipeline looks appropriate. One question I had was, is using the same cost for all amino acid changes and indels in the ancestral sequence reconstruction appropriate here?

Thank you for your positive assessment. To quickly clarify, we added the following to the manuscript:

“Phylogenetic trees are inferred for each country using fasttree⁷¹, and the Sankoff algorithm is applied for ancestral character state reconstruction, and as in Steinbrueck & McHardy, 2011, we use the same cost for all amino acid changes and indels²⁰. We previously showed that in ancestral sequence reconstruction for the major antigen of human influenza A (H3N2) viruses, both approaches produce very similar results, due to the very small evolutionary time scales being considered²⁰.”

Q9: Additionally, perhaps some protein structural analyses could be added to enhance the argument.

This is an excellent point to look at how these structural changes impact the potential binding to antibodies, either monoclonal antibodies or from human sera. We've added a figure (**Fig. 2**; see above) for the antigenic scoring methodology that compares different scoring methods to both antigenic cartography distances as well as the averaged mFRN values (derived from neutralization titers of monoclonal antibodies) for select VOCs.

We added to the discussion following the outlined new analyses:

“In evaluating antigenic alteration predictions, we found that predicted antigenic alteration scores are reflective of how well the evolutionary changes affect the neutralization achieved by antibodies. Studying antigenic scores, acquired mutations and their positioning on the 3D structure might reveal further insights into the mechanistic basis of varying viral variant neutralization.”

Q10: My main concern is with the assessment of antigenic alterations/antigenicity scoring (results on lines 407-441 and methodology on lines 514-522). How are these lineage-specific antigenic alteration scores calculated? What are the thresholds for high vs low antigenic scores? On Line 138 "3.85" was mentioned as a threshold based on being "substantial", but how is this defined? Is this a standardized threshold or is it based on the scores found in this project for the different lineages of Sars-CoV-2? Relying on scores of existing events could potentially be an issue if there is a lot of mutational space that was or was not yet explored by these lineages. I appreciate the difficult problem of reducing false hits while avoiding missing important potential pVOIs, and I think the methodology described here struck an appropriate balance. I think this work is an appropriate addition to the set of tools used for VOI detection but should be taken in context. Lines 391-395 and 443-459 helped elucidate the context and novelty of the work. Largely, I think it would benefit from more details about the methodology and explanations of the specific choices made.

We thank the reviewer for the constructive comments and have addressed this, as suggested. To clarify the details of the antigenic alteration scoring approach we have merged the proposed antigenic alterations manuscript with the CoVerage manuscript as well as added further clarifications to the methods section, as outlined below (Question 10b).

“We quantify the degree of antigenic divergence of a viral isolate relative to another one using an antigenic scoring matrix. The antigenic scoring matrix includes antigenic alteration estimates for pairs of amino acid changes derived from a previously inferred antigenic tree for Influenza A viruses (IAV)²³. These were established by inferring antigenic weights, i.e. antigenic branch lengths, from hemagglutination inhibition (HI) assay data onto a maximum likelihood phylogenetic tree and matching these to branch-specific sets of amino acid changes inferred by ancestral character state reconstruction for the major viral surface protein for human influenza A (H3N2) viruses, combining genetic with evolutionary and serological data (Fig. 7a,b).”

And then later continued in the methods:

“From the inferred IAV antigenic tree, several antigenic alteration matrices specifying weights for all pairs with amino acid changes with evidence in the antigenic tree were defined. The weights serve as a measure for the potential impact that the amino acid substitution has on the antigenicity of the variant. $M_{ant_alt_0}$ includes weights averaging over the respective up or down weights seen for a particular change across all positions, such as the change N_K, assigned a weight of 0.5, averaging over $(0+0+0+2)/4$ (Fig. 7c). Matrix $M_{ant_alt_1}$ includes weights from $M_{ant_alt_0}$ averaged across change and reverse change. The matrix $M_{ant_alt_2}$ includes a total of forty-one weights, and matrix $M_{ant_alt_2}$ includes antigenic weights only for changes occurring at least three times in the antigenic tree. All unspecified pairs of changes in the antigenic matrices include missing values and are not counted.

We then explored whether antigenic weights for pairs of amino acid changes derived from data for the antigenic evolution of human influenza A (H3N2) viruses are predictive for the degree of antigenic evolution of human SARS-CoV-2 viruses. Specifically, the antigenic alteration score for a lineage l is calculated to predict the antigenic alterations of each SARS-CoV-2 spike protein sequence thereof relative to the Wuhan-Hu-1 strain. To this end, antigenic weights from the viral immune escape matrix are applied to the amino acid changes of a lineage throughout the spike protein relative to the Wuhan-Hu-1. The weights are subsequently summed per isolate and averaged across all isolates of a given circulating Pango lineage to calculate the final antigenic score, a_l .

For scoring currently circulating SARS-CoV-2 lineages, we used the metadata provided by GISAID and filtered isolates by the most recent complete month (ie. the previous month, or in the Omicron case study by the desired months) based on the isolate collection date. Sequences under review were removed. Among other information, the metadata contains amino acid substitutions or deletions occurring in each isolate as well as their Pango lineage, which would be used to assign an antigenic weight to each change occurring at the previously defined sites. Once assigned to each change, the individual amino acid weights were then summed per isolate and then averaged across each Pango lineage occurring in the selected month to assign one final antigenic score to each circulating Pango lineage that month, such that:

$$a_l = \frac{1}{n_{il}} \sum w_{aa}$$

where a_l is the antigenic score for the Pango lineage l and n_{il} represents the number of isolates per Pango lineage in the given month and w_{aa} represents the assigned weight for an amino acid change.

In addition to the individual lineage scores, a country-wide antigenic alteration score, a_c , is calculated by summing over all antigenic lineage scores, weighted by their frequencies for a particular period in time in a country c :

$$a_c = \sum \left(a_l \times \frac{n_{ilc}}{n_{ic}} \right)$$

Here, n_{ilc} is the number of isolates of each Pango lineage for country c and n_{ic} is the total number of isolates per country."

Fig. 7: Schematic outline of the technique to predict the antigenic alteration of SARS-CoV-2 spike protein sequences using a IAV-derived viral immune escape matrix. a The inputs for antigenic tree inference are Hemagglutination Inhibition Assay (HIA) data table for IAV and a genealogy inferred from

isolate sequences of the major viral surface protein, and the corresponding amino acid sequences²³ **b** Antigenic tree with antigenic branch weights inferred and amino acid changes in the viral antigenic reconstructed for the evolutionary branches. Values such as '0/2' indicate antigenic up and down branch weights, reflecting the asymmetric nature of hemagglutination inhibition data, where antigenic distance from a viral antigen strain a to antiserum raised against another strain b can be different from the distance of an antiserum raised against antigen a to an antigen b, respectively. The red color traced through the IAV Antigenic Tree indicates an example path between the A1 and A3 variant, for which the sum of edge weights correspond to the red values for A1 - A3 and purple values for A3 - A1 in the HIA table in panel a Several changes in bold are exemplary shown in the IAV Antigenic Tree table, which represents for every position-specific amino acid mapped to a tree branch the antigenic branch weights. The up/down weights inferred in this tree are then compiled into **c** position-agnostic viral immune evasion antigenic alteration matrices for pairs of amino acids, shown here with the example of averaging over inferred up and down weights. **d** These weights are then used to score amino acid changes across the spike protein of SARS-CoV-2 to calculate sequence-specific antigenic alteration scores, which are then combined in lineage and country-wise alteration scores.

Q10b (from above): What are the thresholds for high vs low antigenic scores? On Line 138 "3.85" was mentioned as a threshold based on being "substantial", but how is this defined? Is this a standardized threshold or is it based on the scores found in this project for the different lineages of Sars-CoV-2?

Thank you for pointing this out. To address this comment, we have updated the manuscripts' methods section:

*"To select lineages with significant antigenic variation in comparison to other circulating lineages, each circulating lineage was assigned a z-score based on its monthly antigenic score. To calculate the z-score, the population mean, or the mean of the lineages circulating that month above the frequency threshold was subtracted from the individual lineage score and then divided by the standard deviation of those lineages circulating above the frequency threshold. Lineages with a z-score of greater than or equal to 1 were ultimately selected as antigenically distinct from the other circulating lineages, or within the top 15% of circulating lineages. A z-score of 1 was selected as a threshold as on average WHO defined variants of concern (VOCs) had an average z-score of 1.2 (**Supplementary Fig. 4**)."*

As also demonstrated in **Fig. 3c** below:

Fig. 3: Performance evaluation of CoVerage. **a** The difference in days and frequency of pVOI identification of lineages relative to their peak frequencies. The dots in red indicate predicted antigenically altered pVOIs, while the turquoise dots represent predicted non-altered pVOIs. In the side boxplot, the mean is indicated by the black diamond and the number above it. **b** Antigenic scores of pVOIs identified using data before December 2021. Colors represent different WHO designations. The x-axis shows the month of identification, with "Apr. 20" indicating April 2020. The y-axis displays the $-\log_{10}$ transformed p-value of pVOI detection. The size of the points represents the antigenic score of the pVOI. **c** Standardized (z-score) antigenic scores for variants and their sublineages for different WHO variant categories designations until June 2024. Here the horizontal bar within the box indicates the median, while the black diamond and the number above it represents the mean. "NA" indicates pVOIs not designated by WHO. **d** PRAUC of CoVerage antigenic score and PyR0 R/R_A in classifying/predicting lineages as WHO-designated or non-WHO-designated VUMs, VOIs and VOCs. Data for both methods were collected before December

2021, with the ground truth including WHO-designated lineages up to May 2022 to demonstrate early detection capabilities. In **b** and **c**, "NA" indicates pVOIs not designated by WHO.

Q11: Additionally, I believe the sentence that starts on line 490 has a grammatical error.

Many thanks for the helpful suggestion. We have corrected this to:

“Additionally, we require a significant ($FDR < 0.05$) increase for a certain window and a frequency threshold of 0.1 to report results. The reported date, or the date of the significant increase, is the date of the last sequence in the current window.”

Dear reviewers,

thank you very much for the detailed and constructive comments that you provided for our manuscript entitled "***In silico* genomic surveillance by CoVerge predicts and characterizes SARS-CoV-2 Variants of Interest**". We have carefully reviewed and addressed all comments, which has further improved the manuscript. Specifically, we have introduced the following major changes:

- **Reviewer 1** considered the performance results of different evaluated models to be counter-intuitive. To address this, we added further clarifications to the results section as well as literature based evidence to provide context and explanations for our results.
- **Reviewer 1** voiced concerns regarding the robustness of the validation provided for antigenic alteration predictions. To address this, we provided an additional one-sided Wilcoxon signed-rank test, which is more robust to smaller sample sizes, to compare the selected scoring methodology to the baseline.
- **Reviewer 1** requested more evidence or rationale why patterns in amino acid changes of influenza A viruses associated with alterations in antigenicity in human influenza A viruses should be predictive for antigenic alterations of human SARS-CoV-2 lineages. In response, we provide additional analyses investigating the relationships between the molecular changes and antigenicity weight predictions across these viruses, demonstrating that immune escape in the major antigen of both viruses shares significant similarities in these patterns and that these can be linked to certain biophysical changes.
- **Reviewer 2** reported on difficulties when installing one of the pieces of software. In response, we provided more robust installation mechanisms as a bioconda package and singularity container.
- **Both reviewers** had difficulties with some links provided in the manuscript and the Github repository, which we reformatted for ease of access.

We thank you for your time in reviewing this manuscript and believe that with the revisions provided here we have addressed any remaining concerns.

Yours sincerely,

Alice McHardy and colleagues

Reviewer 1:

Remarks to the Author:

The authors have made substantial revisions to address my major concerns from the previous round, including: (1) better comparison of their work with that of others, (2) enhanced clarity around the methodology for antigenic alteration analysis, (3) revisions to how they report the timing of variant identification with respect to WHO announcements, and (4) a more thorough evaluation of their method's effectiveness, including sensitivity and specificity. I thank the authors for their efforts in addressing these points.

Despite these improvements, I still have a few major concerns about the revised analysis:

1. Validation of antigenic alteration scores

- The improvement in correlation between the authors' antigenic alteration score and experimental data compared to a null model (null $\rho = 0.848$ vs. best model $\rho = 0.888$) appears statistically insignificant given the small dataset (19 variants). With so few data points, this improvement may not be robust.

Thank you for pointing this out. Initially, we calculated the significance of the correlation between the CoVerage antigenic scores method and the experimental data, or antigenic cartography distances (ground truth), resulting in a ρ of 0.89 and a corrected p-value of 1.28×10^{-06} using a Spearman's correlation (corrected for multiple testing using the Benjamini Hochberg FDR, alpha of 0.05) (see page 9, line 222). To further assess whether our prediction aligns significantly closer with the ground truth compared to the baseline prediction, we additionally performed a one-tailed Wilcoxon signed-rank test and have integrated the additional information into the manuscript results (p 9, l. 230):

"With this, we selected the method that applies unidirectional immune escape weights for all amino acid changes across the spike protein (Fig. 2c,d) to progress with, as this method was significantly closer to the antigenic distances in comparison to the baseline method as per a one-sided Wilcoxon signed-rank test on the absolute deviations calculated by subtracting the normalized (using min-max normalization) scoring values from the normalized antigenic cartography distances ($n=19$, p-value 0.0005, Methods, Supplementary Table 3)³¹. Utilizing the normalized deviations allows for the comparison of how closely the tested method correlates to the ground truth antigenic distances in comparison to the baseline method."

- The model performance is also counter-intuitive. For instance, the "weights at antigenic sites" model performs worse than the "weights at all sites" model. However, the manuscript does not clarify which positions are referred to as "defined antigenic sites." This ambiguity needs to be addressed.

We apologize for not stating this more clearly. The defined antigenic sites were obtained from research literature on relevant sites affecting SARS-CoV-2 antigenicity up to June 30, 2021. Ultimately, as the SARS-CoV-2 virus continues to evolve, changes at the initially most relevant sites for immune escape do not accurately reflect the potential antigenicity of the virus. Accordingly, we see a decreasing performance for the weights at antigenic sites model over time. We've clarified and expanded the description of the relevant sections in the manuscript accordingly (p. 9, l. 208):

“We analyzed different ways of scoring the antigenic alterations of lineages based on their changes relative to the original Wuhan strain and assessed their value using experimental data. The evaluated scoring methods include (1) scoring antigenic alterations at amino acid sites of the spike protein using *M_ant_alt_0*, which consists of amino acid sites identified from literature through June 30, 2021 that had a demonstrated capacity to alter the viral antigenic phenotype in the early phase of the pandemic (Supplementary Table 1). These sites were largely identified and characterized from the Alpha, Beta and Delta variants^{28,29}, (2) scoring antigenic alterations of directed changes across all amino acid sites of the spike protein, (3) simply counting amino acid changes that occur at the sites defined in method one, (4) counting amino acid changes across the spike protein, (5) using an alternative bidirectional scoring matrix, *M_ant_alt_1*, with antigenic weights averaged across change and reverse change, respectively on all altered sites for the spike protein, and finally, (6) using an alternative scoring matrix, *M_ant_alt_2*, which included averaged antigenic weights from the Influenza A antigenic tree for amino acid changes that occur at least three times across the tree (Methods).

Overall, each of the methods correlated strongly with both the antigenic distances and averaged mFRN values, which were used as a proxy measure of immune escape capacity in this comparison (Supplementary Table 2). Antigenic alteration scores increased between earlier lineages (Alpha, Beta, Epsilon, Lambda, Delta and Gamma) to Omicron and its subsequent lineages, as Omicron has been proven to be antigenically distinct from prior lineages with an increased immune escape capacity from ancestral variant vaccinated sera³⁰. Only two methods (2 and 5, **Fig. 2c,d** and **Fig. 2i,j**) after a Spearman’s correlation test have a greater correlation with antigenic cartography distances and mFRN values than the baseline method, considering counts of changes across the spike protein (4, **Fig. 2.g,h**; Supplementary Table 2). With this, we selected the method that applies unidirectional immune escape weights for all amino acid changes across the spike protein (**Fig. 2c,d**) to progress with, as this method was significantly closer to the antigenic distances in comparison to the baseline method as per a one-sided Wilcoxon signed-rank test on the absolute deviations calculated by subtracting the normalized (using min-max normalization) scoring values from the normalized antigenic cartography distances (n=19, p-value 0.0005, Methods, Supplementary Table 3)³¹. Utilizing the normalized deviations allows for the comparison of how closely the tested method correlates to the ground truth antigenic distances in comparison to the baseline method. Additionally, the antigenic impact of an amino acid change is not symmetric; for instance, physicochemical properties of the amino acid changes are not the same bidirectionally, and some amino acid changes might alter glycosylation at specific residue sites, while the reverse may not produce the same effect³². This unidirectional effect is more reflected in the chosen model.

Notably, limiting the set of sites to those defined in literature in the earlier stages of the pandemic, was not sufficient to cover the continued antigenic evolution of human SARS-CoV-2 lineages, scoring Omicron and several subsequent lineages, XBB and EG.5, lower than their antigenic cartography distances rank (Fig. 2e,f). The XBB lineage has the N460K amino acid change, which reduces neutralization, but is not scored with this model^{33,34}; the EG.5 lineage carries the F456L change, as also seen in the KP.2 lineage, which increases neutralization resistance and is not scored^{35,36}. Ultimately, the antigenic weights serve as an important addition to the scoring method, while the restriction to known antigenic sites does not improve the scoring.”

And in the Discussion we note:

“By weighing amino acid changes occurring across the entire spike protein, the method accounts for any changes in key antigenic sites as the virus continues to evolve. An initial list of antigenic sites across the spike protein that we compiled with literature through June 30, 2021, was not comprehensive enough to cover the continued antigenic evolution of the virus, as it mostly includes sites identified from the Alpha,

Beta and Delta variants and lacks key antigenic sites such as 59, 460, 456, or 505 that arose in later Omicron lineages such as XBB, EG.5 and XEC³³⁻³⁶.

To clarify which sites are incorporated in the antigenic sites mentioned for matrix M_ant_alt_0, we have also provided a list of these selected sites in the Supplementary Table 1 as shown below:

“Supplementary Table 1: Selected amino acid sites in the SARS-CoV-2 spike protein that have been shown to alter SARS-CoV-2 antigenicity using literature available up to June 30, 2021.”

antigenic_sites	information	Reference
18	escape from NTD-binding mAbs, reduces antibody neutralization	Harvey et al., 2021 (https://doi.org/10.1038/s41579-021-00573-0); McCallum et al., 2021(10.1016/j.cell.2021.03.028)
157	reduces neutralization by mAb (2489)	Harvey et al., 2021 (https://doi.org/10.1038/s41579-021-00573-0); Suryadevara et al., 2021 (10.1016/j.cell.2021.03.029)
234	resistant to neutralizing antibodies	Li et al., 2020 (https://doi.org/10.1016/j.cell.2020.07.012)
417	escape neutralization by mAbs, enhanced binding with ACE2	Starr et al., 2021 (DOI: 10.1126/science.abf9302)
439	increased binding affinity for ACE2 through formation of new salt bridge, resistant to some neutralizing antibodies	Harvey et al., 2021 (https://doi.org/10.1038/s41579-021-00573-0); Li et al., 2021 (https://doi.org/10.1038/s41392-021-00592-6)
440	immune escape; reduced the neutralization by mAbs	Starr et al., 2021 (DOI: 10.1126/science.abf9302)
444	immune escape the neutralization by mAbs and human convalescent sera	Harvey et al., 2021 (https://doi.org/10.1038/s41579-021-00573-0); Liu et al., 2021 (10.1016/j.chom.2021.01.014); Starr et al., 2021 (DOI: 10.1126/science.abf9302)
446	immune escape the neutralization by mAbs and human convalescent sera	Harvey et al., 2021 (https://doi.org/10.1038/s41579-021-00573-0); Liu et al., 2021 (10.1016/j.chom.2021.01.014)
450	immune escape the neutralization by mAbs and human convalescent sera	Liu et al., 2021 (10.1016/j.chom.2021.01.014)
452	reduces neutralization by mAbs, enhances infectivity	Harvey et al., 2021 (https://doi.org/10.1038/s41579-021-00573-0); Li et al., 2020 (https://doi.org/10.1016/j.cell.2020.07.012); Liu et al., 2021 (10.1016/j.chom.2021.01.014)

453	reduces neutralization by mAbs, increase ACE2 affinity	Harvey et al., 2021 (https://doi.org/10.1038/s41579-021-00573-0); Starr et al., 2021 (DOI: 10.1126/science.abf9302)
475	immune escape the neutralization by mAbs and human convalescent sera	Li et al., 2020 (https://doi.org/10.1016/j.cell.2020.07.012)
476	immune escape the neutralization by mAbs	Tortorici et al., 2020 (https://doi.org/10.1126/science.abe3354); Rogers et al., 2020 (https://doi.org/10.1126/science.abc7520)
477	immune escape, resistant to neutralization, enhanced binding with ACE2	Harvey et al., 2021 (https://doi.org/10.1038/s41579-021-00573-0); Liu et al., 2021 (https://doi.org/10.1101/2020.11.06.372037); Liu et al., 2021 (10.1016/j.chom.2021.01.014)
478	immune escape the neutralization by mAbs and human convalescent sera	Liu et al., 2021 (10.1016/j.chom.2021.01.014)
479	immune escape the neutralization by mAbs	Liu et al., 2021 (10.1016/j.chom.2021.01.014)
483	resistant to some neutralizing antibodies and to human convalescent sera	Li et al., 2020 (https://doi.org/10.1016/j.cell.2020.07.012)
484	reduces neutralization of antibodies, immune escape	Harvey et al., 2021 (https://doi.org/10.1038/s41579-021-00573-0); Liu et al., 2021 (10.1016/j.chom.2021.01.014)
486	immune escape the neutralization by mAbs	Liu et al., 2021 (10.1016/j.chom.2021.01.014)
489	immune escape the neutralization by mAbs	Starr et al., 2021 (DOI: 10.1126/science.abf9302)
490	resistant to some neutralizing antibodies	Liu et al., 2021 (10.1016/j.chom.2021.01.014)
493	immune escape the neutralization by mAbs	Liu et al., 2021 (10.1016/j.chom.2021.01.014); Starr et al., 2021 (DOI: 10.1126/science.abf9302)
499	immune escape the neutralization by mAbs and human convalescent sera	Liu et al., 2021 (10.1016/j.chom.2021.01.014)
501	reduces neutralization of RBD antibodies, increases ACE2 binding affinity, increases transmissibility	Harvey et al., 2021 (https://doi.org/10.1038/s41579-021-00573-0); Starr et al., 2021 (DOI: 10.1126/science.abf9302)
681	immune escape the neutralization by mAbs and human convalescent sera	Saito et al., 2021 (https://doi.org/10.1101/2021.06.17.448820)

769	decreased susceptibility to neutralizing antibodies	Harvey et al., 2021 (https://doi.org/10.1038/s41579-021-00573-0)
796	reduction in susceptibility to non-RBD specific antibodies, but decreases infectivity	Kemp et al., 2021 (https://doi.org/10.1038/s41586-021-03291-y)

• **A more fundamental concern remains largely unaddressed: while using patterns of amino acid changes in the hemagglutinin (HA) protein of influenza A as a framework for SARS-CoV-2 spike is an innovative idea, it is unclear why this approach should work given the biological differences between these viruses (e.g., structural and functional differences between HA and spike, glycosylation patterns, antigenic assay differences, and host immune responses). The authors should provide more evidence or rationale for why this framework is applicable.**

Thank you for this valuable suggestion; we have expanded on this, as suggested. We've added these findings to the supplementary material as well as to the "In silico assessment of antigenic lineage alteration" results section:

*"The underlying idea of scoring amino acid changes to predict the antigenic alteration of SARS-Cov-2 lineages is that there are commonalities of how the major antigen of these viral pathogens evades the humoral immune responses evoked by the host, and that this is reflected in antigenic alteration weights for such changes that we inferred from genetic and antigenic data of seasonal influenza A viruses^{10,23}. To evaluate associations between these inferred antigenic alteration weights per amino acid change and the associated physicochemical alterations in charge, a Wilcoxon rank sum test was used. The results showed that charge changes from neutral to positive and the reverse positive to neutral, as well as negative to positive and the reverse positive to negative, were associated with significantly higher antigenic weights compared to changes that did not affect charge (p-values of 0.018 for both charge changes, after Benjamini-Hochberg correction for multiple testing; **Supplementary Fig. 2a**), demonstrating that immune evasion in human influenza A/H3N2 viruses is linked to changes in charge. Indeed, amino acid changes, particularly those involving a positive charge, on the hemagglutinin (HA) protein of Influenza A and the spike protein of SARS-CoV-2 reduce the strength of interactions with neutralizing antibodies and affect binding to host receptors^{24,25}.*

*Amino acid changes assigned large antigenic alteration weights for human influenza A/H3N2 viruses are also preferentially found among antigenicity-altering changes reported for SARS-Cov-2 lineages (Fisher's exact test p-value = 2.88×10^{-16} , Odds Ratio = 15.722, n = 380; these changes referred to can be found here:https://github.com/hzibifo/Corona_Variant_Scoring/blob/main/validation/amino_acid_properties/viral_amino_acid_properties.csv), and these antigenic alteration weights for amino acid changes resulted in an Area under the Curve (AUC) value of 0.802 for detecting known immune escape changes of SARS-Cov-2 (**Supplementary Fig. 2c**), both highlighting a strong relationship in how specific amino acid changes influence antigenicity across both viruses."*

Supplementary Figure 2: a Boxplot of the antigenic alteration weights for amino acid changes in antigenic drift of human influenza A/H3N2 viruses⁶⁶ grouped into different charge categories: neutral / negative ($n = 47$) representing changes from a neutral to a negative amino acid or vice versa, neutral / positive ($n = 60$) representing changes from a neutral to a positive amino acid and vice versa, positive / negative ($n = 10$) representing a change from a positive to a negative amino acid and vice versa, and the unchanged category ($n = 190$), where the amino acid charge stays the same. The FDR corrected p-values (with Benjamini Hochberg procedure) from the Wilcoxon rank sum test comparing the antigenic weights of the amino acid change property groups to the antigenic weights of the unchanged group are shown above the corresponding box and whisker plot. **b** Heatmap of amino acid changes and their antigenic alteration weights used for scoring antigenic alterations of SARS-Cov-2 lineages derived from the influenza A/H3N2 antigenic and genetic data⁶⁶. Dark purple represents a higher weight and a light purple represents a lower weight. Blank spaces in the heatmap represent amino acid changes that do not have a weight. **c** ROC

curve assessing the predictive power of antigenic alteration weights for specific amino acid alterations for changes altering the antigenicity for SARS-CoV-2. The AUC is given in red (AUC of 0.802).

2. Performance evaluation and variant mislabeling

• **Several variants appear mislabelled in Figure 3. For example, in panel b, BA.1 (B.1.1.529.1*) and sublineages of Delta (B.1.617.2*) are labeled as “VUM,” despite being VOCs according to WHO designations before December 2021. This mislabeling has significant downstream impacts on the evaluation of model performance. For instance, the authors claim that 10 of the 12 lineages not captured by their method were VUMs, which are of “lower public health relevance.” Was Omicron BA.1 one of these lineages? This issue undermines the credibility of the performance metrics presented in Figure 3.**

Thank you for noting this. We have adapted the relabelling and reperformed the analysis. The cause of the issue is that the two lineages were redesignated by WHO: B.1.1.529 was initially labelled as a VUM and two days later as a VOC, and B.1.617.2 was first reported as a VOI then five weeks later as a VOC (<https://www.who.int/activities/tracking-SARS-CoV-2-variants>). Instead of using both of their designations, we now only use their final designation to evaluate the precision and recall of predicting these variants overall with pVOIs and for assessing the antigenic scoring as detailed below. This relabeling had only a minor impact on recall (from 73% to 72%) and did not change the precision. Neither of these redesignated lineages were among the 12 pVOI uncaptured lineages. In the updated Fig. 3d, which shows a precision and recall curve of the antigenic scoring, there is also a slight change, accordingly. The AUC for the CoVerage antigenic score is now 0.86 (from 0.84), and for PyR0, it is 0.79 (from 0.74). We have updated all corresponding values throughout the manuscript (page 11, line 290-312).

We also accommodated the lineage relabeling in the revised Fig. 3b-c and separated the subvariants from their parent designated lineages, to demonstrate that CoVerage scoring prioritizes the earliest emerging WHO-designated lineages over their later sublineages. This highlights how the method effectively detects the earliest emerging, antigenically altered lineages, which is crucial for public health decision-making. We integrated this in the revised manuscript as follows:

"The standardized antigenic scores of WHO-designated lineages were much higher than those of the non-designated lineages (Wilcoxon rank sum test p -value= 1.25×10^{-22}), and increasing in order of VUMs, VOIs and VOCs, in line with their proven relevance for public health decision making (Fig. 3c). Notably, CoVerage assigns higher scores to early emerging lineages that exhibit substantial antigenicity changes compared to previously circulating variants, whereas sublineages with minor antigenic alteration from their parent lineages receive relatively lower scores. The scores of subvariants of WHO-designated lineages are lower than those of their parent lineages, although they remain higher than those of non-designated lineages. By prioritizing early emerging lineages with significant antigenic drift, this scoring approach identifies early variants that evade pre-existing immunity and warrant vaccine updates to maintain effective protection"

3. Length and Verbosity

• **The manuscript is lengthy and verbose, especially in the discussion and case studies. While the additional methodological details are appreciated, the authors should aim for conciseness to improve clarity. For example, the discussion could focus on the most critical points, and case studies could be more succinctly presented to highlight key aspects.**

We have addressed this as suggested, shortening the case study sections and the discussion.

Minor Comments

- **Line 48: VOC and WHO abbreviations were already defined on lines 36–37.**

Thank you for noticing this; we have corrected the repetition.

- **Lines 86–87: Clarify what GISAID metadata is referred to (e.g., amino acid substitutions/deletions?). Where in the paper are WHO case numbers used?**

We've clarified what information is referred to in the GISAID metadata:

“Both genomic sequences and the corresponding metadata file, which contain amino acid substitutions and deletions for each given isolate as well as Pango lineage information, are used as inputs for the CoVerage computational analytics pipeline.”

While the WHO case numbers are used in the global map on the sarscoverage.org homepage (Supplementary Fig. 1) which shows WHO reported case numbers, number of available sequences, and the percentage of sequences to cases per country. To clarify this we've added the following section to the results:

“Case numbers are obtained from the WHO Coronavirus (COVID-19) data repository for the global case map on the homepage, which provides an overview of cases and sequences available per country (Supplementary Fig. 1)¹⁹.”

- **Lines 91–93: Rephrase for clarity. The allele dynamics analysis also requires genome sequence data, not just metadata. How does RKI metadata differ from GISAID in this context?**

Thank you for noting this, we've clarified the sentence: *“All of the analyses, the antigenic alteration analysis, the lineage dynamics analysis and the spike protein allele dynamic analysis, require the GISAID metadata as input, though the spike protein allele dynamics also requires genomic sequences.”*

As for the RKI metadata, we've further clarified its difference from the GISAID metadata: *“Optionally, German sequence data is downloaded from Zenodo, where it is made available by the Robert Koch Institute (RKI) (<https://zenodo.org/record/8334829>), prior to its submission to GISAID. Alternative to GISAID metadata, the RKI metadata contains both sequence lineage assignment and submission date information from genomic surveillance efforts within Germany.”*

- **Line 109: What is the added value of performing MSA and ancestral sequence reconstruction for mutation frequency analysis compared to simpler methods like those on platforms such as CoV-Spectrum?**

We added the following to the Methods section:

“The use of ancestral sequences reconstruction allows for the identification of amino acid change events in the evolutionary history of SARS-CoV-2 lineages, which defines lineage-alleles and their frequencies within

a specific time period as the ratio of the number of isolates in the subtree of the allele relative to the number of all isolates within the designated period²⁰. Despite being more complex, this gives a deeper evolutionary understanding of the dynamics of individual lineages and their associated amino acid changes and thus determines whether associated changes increase significantly over time, potentially providing selective advantage²⁰.

- **Line 135: Should reference Figure 7 instead of 6.**

Thank you for pointing this out, we corrected this.

- **Lines 147–148: Asterisks are mentioned in the text but not visible in the heatmaps, only in the lineage dynamics plots. Using asterisks for both antigenic alterations and statistically significant frequency increases is potentially confusing—are their timings distinct?**

The asterisks mentioned here belong to the heatmaps output by the antigenic scoring methodology (ie. those depicted in Fig. 6) not those for the pVOI detection, and thus the timings are distinct. We apologize for this lack of clarity and have adapted the description in the results:

“In the antigenic alteration heatmap, significantly antigenically altered lineages are indicated with an asterisk by the lineage name, otherwise antigenically altered lineages are ranked by frequency.”

And in the legend for Fig. 6:

“Significantly antigenically altered lineages (with a standardized score greater than or equal to 1) are denoted with an asterisk.”

- **Antigenic data source confirmation: Can the authors confirm the data sources (e.g., Table S1 from the Wang et al. preprint for antigenic cartography and Supplementary Data File 2 from the Cox et al. for mean mFRN)?**

Thank you for making note of this, the reviewer is correct. The antigenic cartography distances come from Table S1 in the Wang et al. preprint, however, the mean mFRN values were calculated from Figure 1b in the Cox et al. paper.

In the methods we've add the following to clarify this:

“The geometric mean fold reduction in neutralization (mFRN) values of monoclonal antibodies collected in Cox et al., 2022 were averaged across all Class 1, 2, and 3 mAbs and mAb cocktails presented in Figure 1b of the review, to get the mean mFRN values.”

And:

“The antigenic cartography distances used as a comparison were derived from a merged antigenic map from human and hamster sera neutralization datasets available in Table S1 of the Wang et al. preprint²⁷.”

- **Lines 181–182: Specify which amino acid sites are being referred to. This information is missing from the Zenodo supplementary files as well.**

We have updated the specification for which amino acid sites we are referring to in the figure:

“With antigenic weights at defined antigenic sites (Supplementary Table 1) on the spike protein”

These are the amino acid sites that were curated from literature up through June 30, 2021 that had a demonstrated impact on the antigenic phenotype of SARS-CoV-2. Here we provided them as a Supplementary Table, but can also add them to Zenodo if need be.

- **Line 188 and 769: GitHub links do not work.**

We apologize for this, and believe that the error may have been due to the line number. We've corrected the links.

- **Line 200: The sentence appears incomplete.**

We have expanded upon this sentence based on your revisions about the application of the antigenic weights as shown here:

“Ultimately, the antigenic weights serve as an important addition to the scoring method, while the restriction to known antigenic sites does not improve the scoring.”

- **Line 210: Alpha is not an “early” lineage compared to Beta. Beta emerged earlier in mid-2020 (see <https://www.nature.com/articles/s41579-023-00878-2>).**

Thank you, we've corrected the statement to better reflect this as follows:

“Antigenic alteration scores increased between earlier lineages (Alpha, Beta, Epsilon, Lambda, Delta and Gamma) to Omicron and its subsequent lineages, as Omicron has been proven to be antigenically distinct from prior lineages with an increased immune escape capacity from ancestral variant vaccinated sera³⁰.”

- **Line 273: Could statistical significance (e.g., Wilcoxon test) be tested for differences between designated and non-designated pVOIs?**

Thank you for your suggestion. We performed a Wilcoxon rank sum test, which showed that the WHO-designated lineages, including their sublineages, have a significantly higher score than the non-designated ones ($p\text{-value} = 1.25 \times 10^{-22}$). We added a sentence to the results section as follows:

“The standardized antigenic scores of WHO-designated lineages were substantially higher than those of the non-designated lineages (Wilcoxon rank sum test $p\text{-value}=1.25 \times 10^{-22}$), and increasing in order of VUMs, VOIs and VOCs, in line with their proven relevance for public health decision making (Fig. 3c).”

- **Figure 5: Excellent visualization!**

Thank you very much for the appreciation!

Remarks on code availability:

**For data: The two links provided by the authors appear to be non-functional:
https://github.com/hzi769bifo/corona_lineage_dynamics/tree/main/data and
https://github.com/hzi770bifo/Corona_Variant_Scoring/tree/main/data**

Thank you for checking this carefully and we apologize for this issue. To avoid this, in the revised version we put the links on individual lines.

"GISAID IDs are available at:

https://github.com/hzi-bifo/corona_lineage_dynamics/tree/main/data

https://github.com/hzi-bifo/Corona_Variant_Scoring/tree/main/data "

For code: The first link (https://github.com/hzi-bifo/Corona_Variant_Scoring) is functional, and the code for the analysis is available with clear instructions. However, the second link (https://github.com/hzi-775bifo/corona_protein_dynamics) does not appear to be working.

These links included the line numbers from the pdf, since they were spread across them. To avoid this, in the revised version we put the links on individual lines.

"All the codes are publicly available at:

https://github.com/hzi-bifo/corona_lineage_dynamics

https://github.com/hzi-bifo/Corona_Variant_Scoring

https://github.com/hzi-bifo/corona_protein_dynamics"

Reviewer 2

Remarks to the Author:

I believe the authors appropriately addressed my comments and concerns. I hope that this work will help identify potential VOIs.

Thank you for all of your thoughtful suggestions.

Remarks on code availability:

The link in the README file (https://github.com/hzi-bifo/Corona_Variant_Scoring/blob/main/README.md) to the BIFO servers is broken in the following quote...

"Required Environment

The analysis pipeline is designed to run on BIFO servers, specifically along with the Sarscoverage environment that can be found here: https://github.com/hzi-bifo/coverage/tree/main/generate_web_data "

Thank you for your careful assessment of our codes. These sections were only relevant for in-house installation and have been removed from the README for clearer instruction (https://github.com/hzi-bifo/Corona_Variant_Scoring/tree/main).

I downloaded and attempted to test https://github.com/hzi-bifo/corona_lineage_dynamics using the directions in the README file on the simulated data but unfortunately, I ran into some issues with dependencies on my machine.

Thank you very much for the careful assessment of our codes. We have reproduced this issue and resolved it by including necessary R package dependencies into the Conda environment definition file (environment.yml). Additionally, we have introduced two more robust installation methods:

1. A Conda package, *corona_lineage_dynamics*, now available in the *bioconda* channel. This package can be installed along with its dependencies using the command (assuming Conda is already installed on the system):

```
conda install bioconda:corona_lineage_dynamics
```

2. A Singularity container, which requires only Singularity to be installed on the system, eliminating the need for additional dependencies.

All three installation options, along with detailed instructions on installation and testing, are provided in the GitHub repository: https://github.com/hzi-bifo/corona_lineage_dynamics.

Dear reviewer 1,

thank you very much for the detailed feedback for the code provided in our manuscript entitled “***In silico* genomic surveillance by CoVerage predicts and characterizes SARS-CoV-2 Variants of Interest**”. We have carefully reviewed the respective GitHub repositories and have addressed the remaining concern as outlined below.

We sincerely thank you for the time you have taken to carefully review our manuscript and code. We are confident that with the additional revisions provided, we have addressed these issues.

Yours sincerely,

Alice McHardy and colleagues

Reviewer 1

Remarks to the Author:

I have no further comments.

Remarks on code availability:

Regarding the code, I initially only checked that the necessary documents were present, but I have now attempted to run the pipelines for both `corona_lineage_dynamics` and `S-protein_dynamics`. I encountered errors in both:

• **`corona_lineage_dynamics`: Running `SDPlots_lineages_local.sh` resulted in a connection timeout error when attempting to retrieve alias and lineage date files via R (`get_alias_list.R`):**

```
Error in funcon(type, msg, asError = TRUE) :  
Connection timed out after 300040 milliseconds  
Calls: getURL -> curlPerform -> -> fun  
Execution halted
```

• **`corona_protein_dynamics`: The provided `environment.yml` file for setting up the conda environment resulted in a “`ResolvePackageNotFound`” error, preventing installation:**

```
conda env create -f environment.yml  
Collecting package metadata (repodata.json): done  
Solving environment: failed  
ResolvePackageNotFound
```

It would be helpful if the authors could provide clearer instructions for running the code, particularly ensuring all dependencies are properly defined and any external data sources are accessible. That said, I did verify that the actual outputs displayed on their website (`sarscoverage.org`) appear to be functioning.

Thank you for thoroughly testing the code. We apologize for the installation issues you encountered. After extensive testing, we identified that:

1. The "Connection timed out" error in the "`corona_lineage_dynamics`" pipeline is caused by internet connectivity problems when downloading lineage alias data from `cov-lineages.org`. To address this, we have provided detailed instructions on how to manually obtain this file in case of connection issues. Additionally, we have included a pre-downloaded file in the test folder. Thus, when the website `cov-lineages.org` is not accessible, the test pipeline automatically uses the pre-downloaded file.
2. The “`ResolvePackageNotFound`” error occurs because the "defaults" channel was removed from the `environment.yml` file, as it now requires a license. Certain package versions/builds are

exclusively available in the defaults channel. We have updated the YAML file to source packages only from the "bioconda" and "conda-forge" channels. As mentioned in the README, besides using Conda for installation, users can also use the provided Singularity image, which is encouraged for its better reproducibility.

Furthermore, we have updated the README file to include more detailed installation instructions, input and output file descriptions.